# Mixed representations of choice direction and outcome by GABA/glutamate cotransmitting neurons in the entopeduncular nucleus

Julianna Locantore[1], Yijun Liu[1], Jesse White[1], Janet Berrios Wallace[2], Celia Beron[2], Emily Kraft[1], Bernardo Sabatini[2], Michael Wallace[1]*

[1]Department of Anatomy and Neurobiology, Boston University Chobanian & Avedisian School of Medicine, Boston, United States; [2]Howard Hughes Medical Institute, Department of Neurobiology, Harvard Medical School, Boston, United States

*For correspondence:
mlwall12@bu.edu

Competing interest: The authors declare that no competing interests exist.

## eLife Assessment

Somatostatin-expressing neurons of the entopeduncular nucleus (EPNSst+) provide a limbic output of the basal ganglia and co-release GABA and Glutamate in their projection to the lateral habenula, a structure that is key for reward-based learning. Combining fiber photometry and computational modeling, the authors provide **compelling** evidence that EPNSst+ neural activity represents movement, choice direction and reward outcomes in a probabilistic switching task but, surprisingly, neither chronic genetic silencing of these neurons nor selectively elimination glutamate release affected behavioral performance in well-trained animals. This **valuable** study shows that despite its representation of key task variables, EPNSst+ neurons are dispensable for ongoing performance in a task requiring outcome monitoring to optimize reward. This work will be of interest to those interested in neural circuits, learning, and/or decision making.

**Abstract** The basal ganglia (BG) are an evolutionarily conserved and phylogenetically old set of sub-cortical nuclei that guide action selection, evaluation, and reinforcement. The entopeduncular nucleus (EP) is a major BG output nucleus that contains a population of GABA/glutamate cotransmitting neurons (EP$^{Sst+}$) that specifically target the lateral habenula (LHb) and whose function in behavior remains mysterious. Here, we use a probabilistic switching task that requires an animal to maintain flexible relationships between action selection and evaluation to examine when and how GABA/glutamate cotransmitting neurons contribute to behavior. We find that EP$^{Sst+}$ neurons are strongly engaged during this task and show bidirectional changes in activity during the choice and outcome periods of a trial. We then tested the effects of either permanently blocking cotransmission or modifying the GABA/glutamate ratio on behavior in well-trained animals. Neither manipulation produced detectable changes in behavior despite significant changes in synaptic transmission in the LHb, demonstrating that the outputs of these neurons are not required for ongoing action-outcome updating in a probabilistic switching task.

## Introduction

Animals select actions based on incoming sensory information, their current state, and past experience to achieve goals. Experiences modify behavior to promote the repetition of actions associated

with positive outcomes and suppress those associated with bad outcomes. Nevertheless, it is also advantageous to maintain flexibility and adjust behavior if the environment changes and to exploit new opportunities as they arise. The BG are an evolutionarily ancient group of nuclei in the brain conserved in all vertebrates and crucial for goal-directed movements, including behavioral updating as a consequence of experience (*Graybiel et al., 1994*). The BG is involved in both action repetition and exploration and consequently, defects in the BG contribute to human disorders ranging from Parkinson's and Huntington's disease, to drug addiction (*Hyman et al., 2006*; *Nelson and Kreitzer, 2014*). Despite the importance of BG to human behavior, how these evolutionarily conserved and phylogenetically old nuclei carry out these functions is not fully understood.

Neural activity from many areas of sensory, motor, and limbic cortices converge onto the striatum, the main input structure of the BG (*Gerfen, 1992*). The dorsal striatum modulates the output nuclei of the BG, the substantia nigra reticulata (SNR), and the EP, through two routes. The so-called direct pathway is formed by dopamine receptor type 1 (D1R) expressing striatal projection neurons (SPNs) that synapse onto output neurons in the SN and EP. The indirect pathway consists of dopamine receptor type 2 (D2R) expressing SPNs that innervate the globus pallidus externus (GPe), which projects to the SNR and EP. GPe also innervates the subthalamic nucleus (STN) which projects to SNR/EP. Canonically, the SNR and EP modulate motor output through their connections to cortically-projecting thalamic nuclei (*Nelson and Kreitzer, 2014*; *Gerfen, 1992*). The function of EP is of particular interest because, whereas the EP in rodents (and globus pallidus internus (GPi) in primates) clearly has motor functions, it is distinct from SNR in that it projects to the LHb and carries reward and sensory signals, implying additional limbic and associative functions (*Hong and Hikosaka, 2008*; *Parent and De Bellefeuille, 1982*). Our previous work demonstrated that the mouse EP has at least two genetically defined cell-types that project to the LHb, one expresses Parvalbumin (*Pvalb*) and vGluT2 (*Slc17a6*) and is purely glutamatergic (excitatory), the other expresses Somatostatin (*Sst*), vGluT2 (*Slc17a6*), and vGaT (*Slc32a1*) and cotransmits both GABA and glutamate (*Wallace et al., 2017*; *Kim et al., 2022*). Here, we focus on activity patterns of cotransmitting $EP^{Sst+}$ neurons in freely moving behavior and their involvement in ongoing action selection and outcome evaluation.

Cotransmitting neurons are increasingly recognized as important contributors to neural circuit function throughout the brain, but specifically manipulating one transmitter at a time to assess impacts on behavior has been challenging (*Wallace and Sabatini, 2023*; *Yao et al., 2023*). To determine the function of *Sst*+ GABA/glutamate cotransmitting EP neurons during behavior, we developed a probabilistic switching task where animals choose between two nose-poke ports that asymmetrically and probabilistically deliver water rewards. The task alternates the location (left or right) of the highly rewarded port every 50 rewards (referred to as a block transition), requiring the animal to remain flexible to maximize the number of rewards it receives. We find that animals are sensitive to changes in reward probability and accurately follow the location of the highly rewarded port following a block transition. $EP^{Sst+}$ neurons are strongly engaged during this task and show bidirectional changes in activity during the choice (i.e. ipsiversive vs contraversive movements towards the side ports) and outcome periods of the task. We then test the requirement for ongoing cotransmission of both GABA and glutamate from $EP^{Sst+}$ to the LHb for continued task performance. Additionally, we alter the GABA/glutamate ratio of cotransmission by genetically deleting vGluT2 from $EP^{Sst+}$ neurons. Despite observing strong modulation of their activity during a trial, neither manipulation of synaptic release resulted in detectable changes in task performance and we conclude that $EP^{Sst+}$ neurons are not required for ongoing trial-to-trial action-outcome evaluation in well-trained animals as assessed on a probabilistic switching task.

## Results

### Changing reward probabilities affects performance on a probabilistic switching task

To examine the activities of $EP^{Sst+}$ cotransmitting neurons during behavior we employed a dynamic, probabilistic switching task in mice modeled after behavioral paradigms shown to require basal ganglia circuitry (*Stephenson-Jones et al., 2016*; *Tai et al., 2012*; *Hamid et al., 2016*; *Chantranupong et al., 2023*). Water-restricted, freely-moving animals are placed in a behavioral arena with three nose-poke ports. A center poke initiates a trial, then the animal chooses to poke the left or right-side ports to

receive a water reward (~3 µL, *Figure 1a*). Water rewards are delivered asymmetrically and probabilistically in a block structure such that once 50 rewards are gained, the reward probabilities are reversed (referred to as a block transition, *Figure 1b*, dotted vertical line). There is no cue that a transition to a new block has occurred; therefore, following a block transition, the probability that the animal chooses the highly rewarded port (*p*(high port)) drops dramatically. As the animal adjusts its choices, *p*(high port) gradually increases for the next 10–15 trials (*Figure 1d*). A well-trained animal will use its history of choices (left or right) and outcomes (rewarded or unrewarded) to guide future actions and is sensitive to block transitions (*Figure 1b–e*). Lights above the ports indicate when the center or side ports are active to assist training, but otherwise provide no information regarding the location of the highly rewarded port. Well-trained animals perform at least 350–500 trials in a 40 min session with individual trials totaling 2–3 s with an enforced minimum inter-trial interval of 1 s (*Figure 1a-c*, *Figure 1—figure supplement 1g*). The trial typically begins with a quick center port entry and exit, after which the animal must poke to the left or right-side port within 8 s and lick the water spout (*Figure 1c*). If a reward is delivered the animal continues to lick and consume the reward (~1 s). If the trial is unrewarded, the animal quickly returns to the center port to initiate a new trial (*Figure 1c*).

To test how different reward probabilities affected task performance we chose three pairs of reward probabilities ranging from more deterministic (90/10) to more stochastic (70/30) (*Figure 1*, red:90/10, blue:80/20, and green:70/30). We observed strong effects on several behavioral metrics (*Figure 1d–i*). First, following a block transition, *p*(high port) returned to pre-block transition levels more quickly (fewer trials) with 90/10 reward probabilities than with 80/20 or 70/30 (*Figure 1d*). This was quantified by fitting an exponential to the first 20 trials following a block transition and extracting the time constant tau$_{p(high\ port)}$, *Figure 1d* inset, and (*Figure 1—figure supplement 1a*). Second, the probability of the mouse switching side port choice on consecutive trials (p switch) increased sharply following a block transition, and the maximum *p*(switch) was greatest and increased the fastest for 90/10 reward probabilities (*Figure 1e* and *Figure 1—figure supplement 1b*). Together, these measures indicate that the animals adapt their behavior most rapidly under these conditions. Consequently, across all trials, the probability that a trial is rewarded (*p* reward) is greatest in the 90/10 reward probability (mean:0.78, SEM:0.001), and progressively decreases with 80/20 and 70/30 conditions (*Figure 1g*). Given that a block transition occurs only after 50 rewards are gained, decreased *p*(reward) results in increased block lengths for 80/20 and 70/30 conditions relative to the 90/10 condition (*Figure 1—figure supplement 1i*).

Unlike *p*(reward) and *p*(high port), the average *p*(switch) across all trials does not depend on the reward probability conditions (*Figure 1f–h*). However, differences in switching behavior are revealed when trials are separated by the outcome of the previous trial. Across conditions, *p*(switch) following a rewarded trial is low (*Figure 1i*; 90/10: p(switch)=0.04), and following an unrewarded trial is high (*Figure 1j*; 90/10: *p*(switch)=0.35). However, the 70/30 condition shows *p*(switch) significantly lower than the 90/10 condition following both rewarded and unrewarded trials (*Figure 1i–j*). Therefore, the large impact changing reward probabilities has on *p*(switch) is revealed by considering the outcome of the previous trial (*Figure 1e–j*). But when considering all trials (*Figure 1h*), there is no difference in *p*(switch) across reward probabilities simply because there are far fewer rewarded trials in the 70/30 condition (*Figure 1g*). These findings prompted us to detail the impact of outcome and choice on *p*(switch) by conditioning this probability on all possible combinations of trial history for two trials in the past (*Figure 1—figure supplement 1d–e*). For almost all types of trial history, the 70/30 condition had the lowest *p*(switch) indicating that the animal is much less likely to switch ports following the most action-outcome pairings in the 70/30 condition (*Figure 1—figure supplement 1e*) which likely contributes to decreased overall *p*(high port).

Finally, we used a linear model (termed Recursively Formulated Logistic Regression; RFLR), previously developed to describe the behavior of a mouse performing the probabilistic switching task, to examine if the animal's strategy changed with different reward probabilities (*Chantranupong et al., 2023*; *Beron et al., 2022*). In this model, the next choice is based on evidence about the location of rewards, represented by the interaction between choice (left or right) and outcome (rewarded or unrewarded; *Figure 1—figure supplement 2a*). A parameter (β) captures the contribution of new evidence from each trial's choice and outcome and the exponentially decaying influence of previous (past trials) evidence on the mouse's next choice. The model includes an additional bias towards or away from its most recent choice (*Figure 1—figure supplement 2a*). Thus, the parameters for the RFLR capture

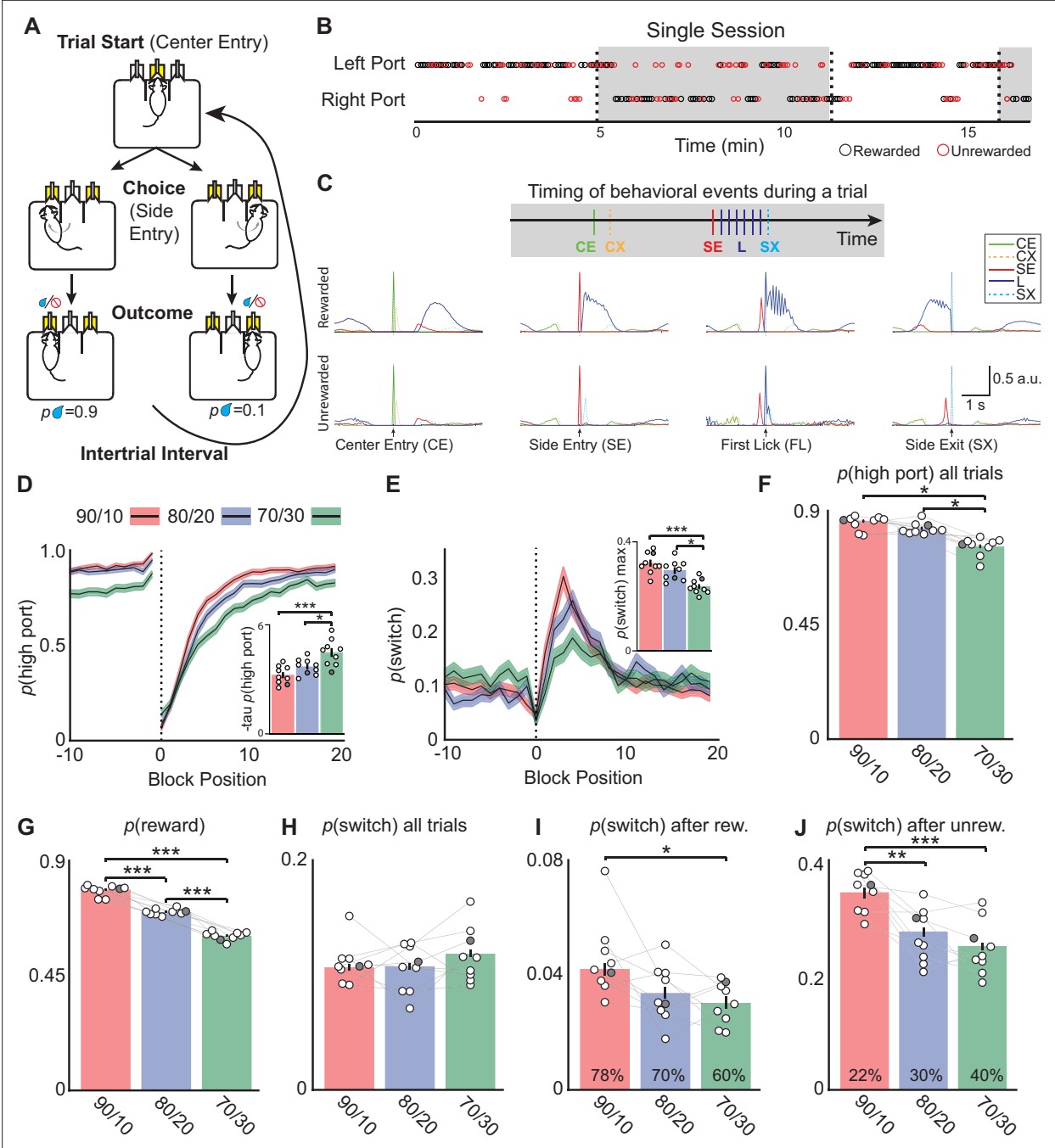

**Figure 1.** Mice alter choices to changing reward probabilities in a probabilistic switching task. (**A**) Illustration of the animal movements and epochs (Trial Start, Choice, and Evaluation) of a single trial in the probabilistic two-port choice task. Yellow color of port(s) indicates when LED is active during a trial. (**B**) A sample of a behavioral session showing periods when the highly rewarded port is on the left (white) and when it switches to the right (gray). The reward probabilities switch (dotted vertical lines, 'block transition') once 50 rewards are gained by the animal. Rewarded trials are represented by black circles and unrewarded trials are red circles, reward probabilities are 70/30. (**C**) Probability distributions of different behavioral events during rewarded (top) and unrewarded (bottom) trials to illustrate the timing of different events within a trial (one session ~500 trials, 90/10 rew. prob). CE = Center Entry, CX = Center Exit, SE = Side Entry, FL = First Lick, SX = Side Exit. (**D**) The probability of choosing the highly rewarded port ($p$high port) around a block transition (dotted vertical line) for different reward probabilities (black line = mean, shaded area = SEM). (Inset) Bar plot showing tau$_{p(\text{high port})}$ (time constant) calculated from an exponential fit to the first 20 trials following a block transition for each animal (circles) (bar = mean, error bar = SEM) in the different reward probabilities. The darkened circle represents the same animal across reward probabilities (see **D-J**) (one-way ANOVA df = 24, p-values: 90/10 vs 80/20=0.22, 90/10 vs 70/30=0.001, 80/20 vs 70/30=0.02). (**E**) The probability of choosing different side ports on consecutive trials ($p$ switch) around a block transition (dotted vertical line) for different reward probabilities (black line = mean, shaded area = SEM). (Inset) Bar plot showing

*Figure 1 continued on next page*

*Figure 1 continued*

the maximum p(switch) in the 20 trials that follow a block transition for each animal (circles) (bar = mean, error bar = SEM) in the different reward probabilities (one-way ANOVA df = 24, p-values: 90/10 vs 80/20=0.23, 90/10 vs 70/30<0.001, 80/20 vs 70/30=0.003). (**F**) The probability of choosing the highly rewarded port on all trials across reward probabilities (bar = mean, error bar = SEM) (one-way ANOVA df = 24, p-values: 90/10 vs 80/20=0.25, 90/10 vs 70/30<0.001, 80/20 vs 70/30<0.001). (**G**) The probability that a trial results in a reward across reward probabilities (bar = mean, error bar = SEM) (one-way ANOVA df = 24, p-values: 90/10 vs 80/20<0.001, 90/10 vs 70/30<0.001, 80/20 vs 70/30<0.001). (**H**) p(switch) across all trials for different reward probabilities (bar = mean, error bar = SEM). (**I**) p(switch) for trials following a rewarded trial for different reward probabilities, percentages in bars represent the proportion of rewarded trials for each condition, also shown in (**G**) (bar = mean, error bar = SEM) (one-way ANOVA df = 24, p-values: 90/10 vs 80/20=0.06, 90/10 vs 70/30=0.01, 80/20 vs 70/30=0.74). (**J**) p(switch) for trials following an unrewarded trial for different reward probabilities, percentages in bars represent the proportion of unrewarded trials for each condition (bar = mean, error bar = SEM) (one-way ANOVA df = 24, p-values: 90/10 vs 80/20=0.002, 90/10 vs 70/30<0.001, 80/20 vs 70/30=0.55). For (**D–J**) n=9 male mice, 8–10 sessions/mouse/rew. prob, ~550 trials/session, total # of trials per condition are: 90/10: 44,512, 80/20: 42,302, 70/30: 43,796.

The online version of this article includes the following figure supplement(s) for figure 1:

**Figure supplement 1.** Behavioral data showing individual animals.

**Figure supplement 2.** Behavioral modeling using a recursively formulated logistic regression (RFLR).

the tendency of an animal to repeat its last choice (alpha, α), the weight given to evidence about past choice and outcome (beta, β), and the time constant (tau, $\tau$) over which the influence of choice and outcome history changes (*Figure 1—figure supplement 2a–c*). Importantly, the model performed equally well across reward probabilities as measured by the negative log-likelihood of the fit (90/10 = −0.25 SD = 0.03; 80/20 = −0.24 SD = 0.03; 70/30 = −0.25 SD = 0.02) and accurately predicted mouse behavior p(switch) and p(high port) around block transitions (*Figure 1—figure supplement 2d–f*). Modeling the different reward probabilities revealed that the most stochastic reward probability (70/30) had the greatest beta (β) and tau ($\tau$) coefficients indicating that to accurately represent the animal's behavior the model needed to use evidence (from the previous choice and outcome) accumulated from trials further in the past (*Figure 1—figure supplement 2b–c*). This strategy likely arises in conditions in which rewarded outcomes are more random (such as the 70/30 condition) and accounting for more past trials can improve the animal's chances of determining the location of the highly rewarded port. Using the RFLR we created psychometric curves of the animal's choice behavior by comparing the log-odds ($\Psi_{t+1}$) calculated from the RFLR to the animal's actual choices p(left port chosen); (*Figure 1—figure supplement 2g–h*). In agreement with other reports (*Stephenson-Jones et al., 2016*; *Tai et al., 2012*), the data was well fit by a logistic function and different reward probabilities were better fit by different functions than the same function (*Figure 1—figure supplement 2h*, p=0.002). The 70/30 reward probability had the steepest slope, indicating that in this condition the animal continues to choose the same port as the previous trial despite increasing evidence to switch to the other side and is consistent with our observations regarding decreased p(switch) (*Figure 1i–j*).

## EP^Sst+ neurons respond during the choice and outcome phases of a trial

To examine the activity of projection and genetically defined neuronal populations in the EP, we injected Cre-dependent GCaMP6f and tdTomato into EP and AAVretro Flp-dependent Cre into the LHb of a *Sst*-Flp mouse line. This resulted in GCaMP6f and tdTomato expression specifically in the *Sst* + neurons of the EP without off-target expression in surrounding areas (*Figure 2a*). We implanted a fiber optic above the EP and recorded EP^Sst+ population calcium-mediated fluorescence changes during the probabilistic switching task using fiber photometry. We consistently observed dynamic changes in GCaMP6f mediated fluorescence while the animal was engaged in the task that was not present in the control (tdTomato) static fluorophore (*Figure 2b* and *Figure 2—figure supplement 1a-c*). We aligned photometry signals to different behavioral events to examine how the EP^Sst+ activity changed relative to different periods of a trial (*Figure 2c*). When we segregated trials by the direction the animal made its side-port choice (ipsi = same side as the recording site, contra = opposite side to the recording site) we observed large differences in EP^Sst+ neuronal activity (*Figure 2c*). Following the center port entry (CE) a large rise in fluorescence was present in ipsilateral trials not seen in contralateral trials (*Figure 2c and e*). This increase in activity was seen for all three reward probabilities tested (90/10, 80/20, and 70/30) and occurred while the animal was engaged in ipsiversive movements during the 'choice' phase of a trial as the animal moved towards the ipsilateral side port (*Figure 2c and e*). The animal also made ipsiversive movements at other phases of a trial such as when it returned

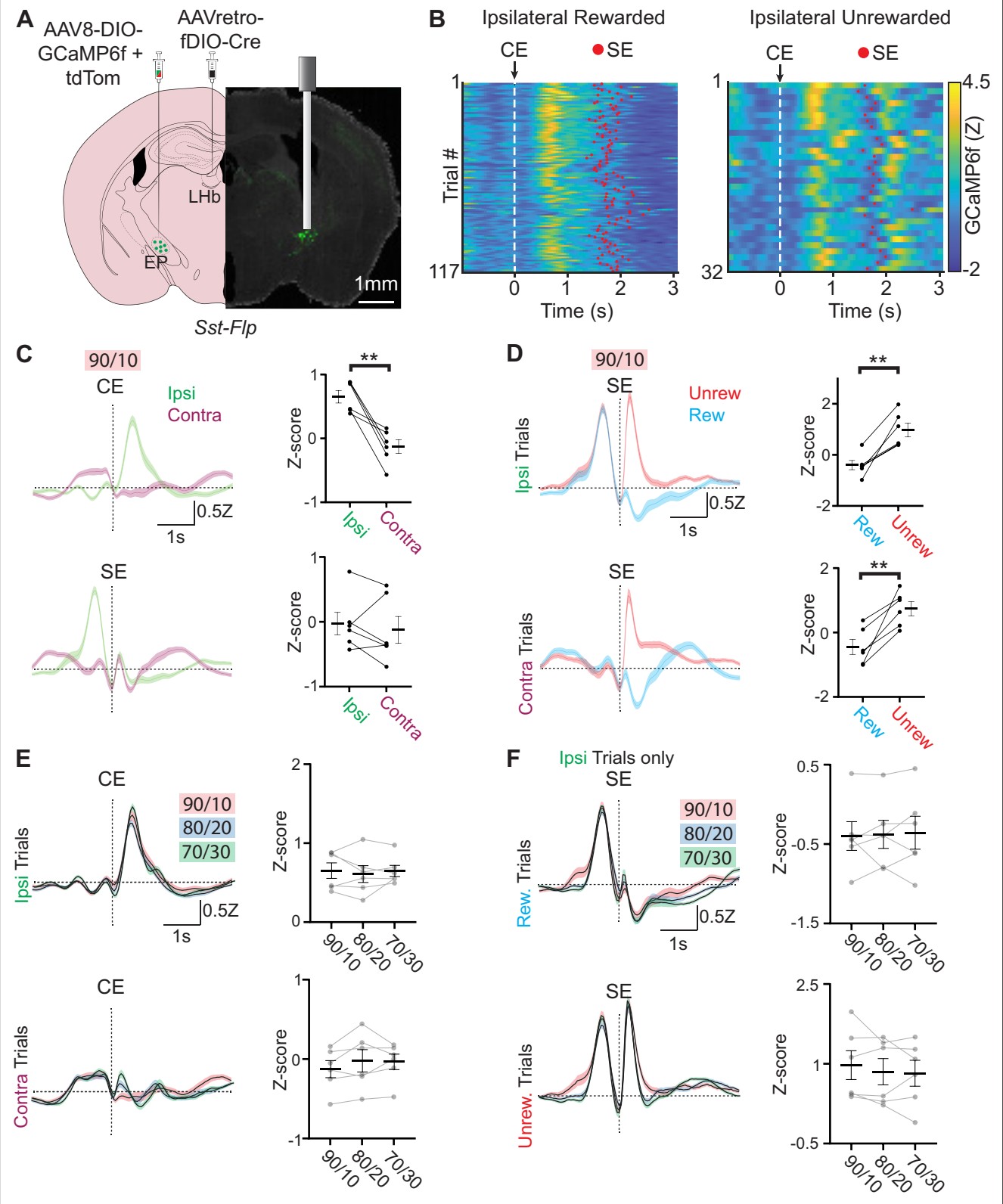

**Figure 2.** Neural activity in EP^Sst+ neurons encodes both choice and value. (**A**) Viral injection location for specific infection of EP^Sst+ neurons with GCaMP6f in a *Sst-Flp* mouse line and fiber implant location for photometry recording. (**B**) Fiber photometry recording of EP^Sst+ neurons for individual trials during a behavioral session. Trials are aligned to center port entry (CE) and red dots indicated side port entry (SE). Only trials to the ipsilateral side (relative to the photometry recording) are shown and are divided by rewarded (left) and unrewarded (right) trials. (**C**) Averaged (± SEM) photometry

*Figure 2 continued on next page*

*Figure 2 continued*

signals across all mice aligned to center-port entry (CE, top) or side-port entry (SE, bottom) grouped by ipsilateral (green) and contralateral (magenta) choice, dotted horizontal line represents z-score equal to zero (all sessions are at 90/10 reward probability, n=6 male mice, 49 sessions, 20,355 trials). (right) Points represent the mean z-scored fluorescence per animal for the 500ms period immediately following the behavioral event, bars represent mean across animals ± SEM (all sessions are at 90/10 reward probability, n=6 male mice, 49 sessions, 20,355 trials) (top; paired t-test DF = 5, p=0.003). (**D**) Averaged photometry (± SEM) signals across all mice aligned to side port entry (SE) grouped by rewarded (blue) or unrewarded (red) outcomes and divided by ipsilateral choice (top) or contralateral choice (bottom). (right) Points represent the mean z-scored fluorescence per animal for the 500ms period immediately following the behavioral event, bars represent mean across animals ± SEM (all sessions are at 90/10 reward probability, n=6 male mice, 49 sessions, 20,355 trials) (top; paired t-test DF = 5, p=0.003, bottom paired t-test DF = 5, p=0.006). (**E**) Averaged (± SEM) photometry signals across different reward probabilities aligned to center port entry (CE) and divided by ipsilateral (top) and contralateral (bottom) choice. (right) Points represent the mean z-scored fluorescence per animal for the 500 ms period immediately following the behavioral event, bars represent mean across animals ± SEM (90/10: n=6 male mice, 49 sessions, 20,355 trials, 80/20: n=6 male mice, 54 sessions, 27,433 trials, 70/30: n=6 male mice, 49 sessions, 27,174 trials). (**F**) Averaged (± SEM) photometry signals across different reward probabilities aligned to side port entry (SE) and divided by rewarded (top) and unrewarded (bottom) outcomes. (right) Points represent the mean z-scored fluorescence per animal for the 500 ms period immediately following the behavioral event, bars represent mean across animals ± SEM (90/10: n=6 male mice, 49 sessions, 20,355 trials, 80/20: n=6 male mice, 54 sessions, 27,433 trials, 70/30: n=6 male mice, 49 sessions, 27,174 trials).

The online version of this article includes the following figure supplement(s) for figure 2:

**Figure supplement 1.** Alignment and sorting of photometry signals from EP$^{Sst+}$ neurons to different behavioral events.

**Figure supplement 2.** Examining the influence of reward history on photometry signals from EP$^{Sst+}$ neurons.

to the center port following a contralateral choice (e.g. SX$_{Contra}$ to CE; *Figure 2—figure supplement 1f*). We observed an increase in calcium activity during these ipsiversive movements (e.g. SX$_{Contra}$ to CE) as well, but they were not as large as those observed during the choice phase (*Figure 2—figure supplement 1f*). Therefore, during the choice phase of a trial, activity contains signals related to directional movement and possibly additional factors.

A large increase in EP$^{Sst+}$ neuronal activity was also observed following side port entry (SE) on unrewarded trials for both contralateral and ipsilateral choices (*Figure 2d*). This was mirrored by a distinct decrease in fluorescence on rewarded trials following side port entry (*Figure 2d*). Increased EP$^{Sst+}$ neuronal activity following an unrewarded outcome was partially due to the rapid withdrawal of the animal's snout following an unrewarded outcome, however, differences in rewarded and unrewarded trials were still distinguishable when signals were aligned to side port exit indicating that these increases in EP$^{Sst+}$ neuronal activity on unrewarded trials were a combination of outcome evaluation (unrewarded) and side port withdrawal occurring in quick succession (SX, *Figure 2—figure supplement 1e*).

One hypothesis is that these outcome signals reflect reward prediction error (*Stephenson-Jones et al., 2016*), which implicitly reflects expectation (given that reward size does not change trial-to-trial). Under different reward probability conditions, the expected reward and corresponding error should scale; however, these patterns in response to rewarded and unrewarded trial outcomes were virtually identical on all reward probabilities tested (*Figure 2e–f*) indicating that they are unlikely to reflect changes in reward expectation (however, see *Tobler et al., 2005*). To further examine if reward prediction error (RPE) contributed to the changes in EP$^{Sst+}$ neuronal activity observed following side port entry, we divided trials by whether the previous trial (trial$_{-1}$) was rewarded or unrewarded. For rewarded trials (both ipsilateral and contralateral), we observed a small effect of the previous trial outcome on EP$^{Sst+}$ activity following side port entry (SE) (*Figure 2—figure supplement 2c–d*). EP$^{Sst+}$ activity on rewarded trials was increased when the previous trial was unrewarded, however, this effect of trial history was not observed on unrewarded trials (*Figure 2—figure supplement 2c–d*). Therefore, the bidirectional changes in EP$^{Sst+}$ neuronal activity observed during the action evaluation period of a trial likely reflect a combination of outcome value and differential timing of movement sequences on rewarded and unrewarded outcomes. In sum, we saw two timepoints with differential activity during a trial: at trial initiation (CE), we found increased activity specifically during ipsiversive movements. Then, during outcome evaluation (SE), we found bidirectional modulation dependent on reward outcome.

## Direction and outcome shape EP$^{Sst+}$ activity

The movements of an animal during a trial of the probabilistic switching task are complex and occur in quick succession making it difficult to disambiguate which behavioral events may be associated with

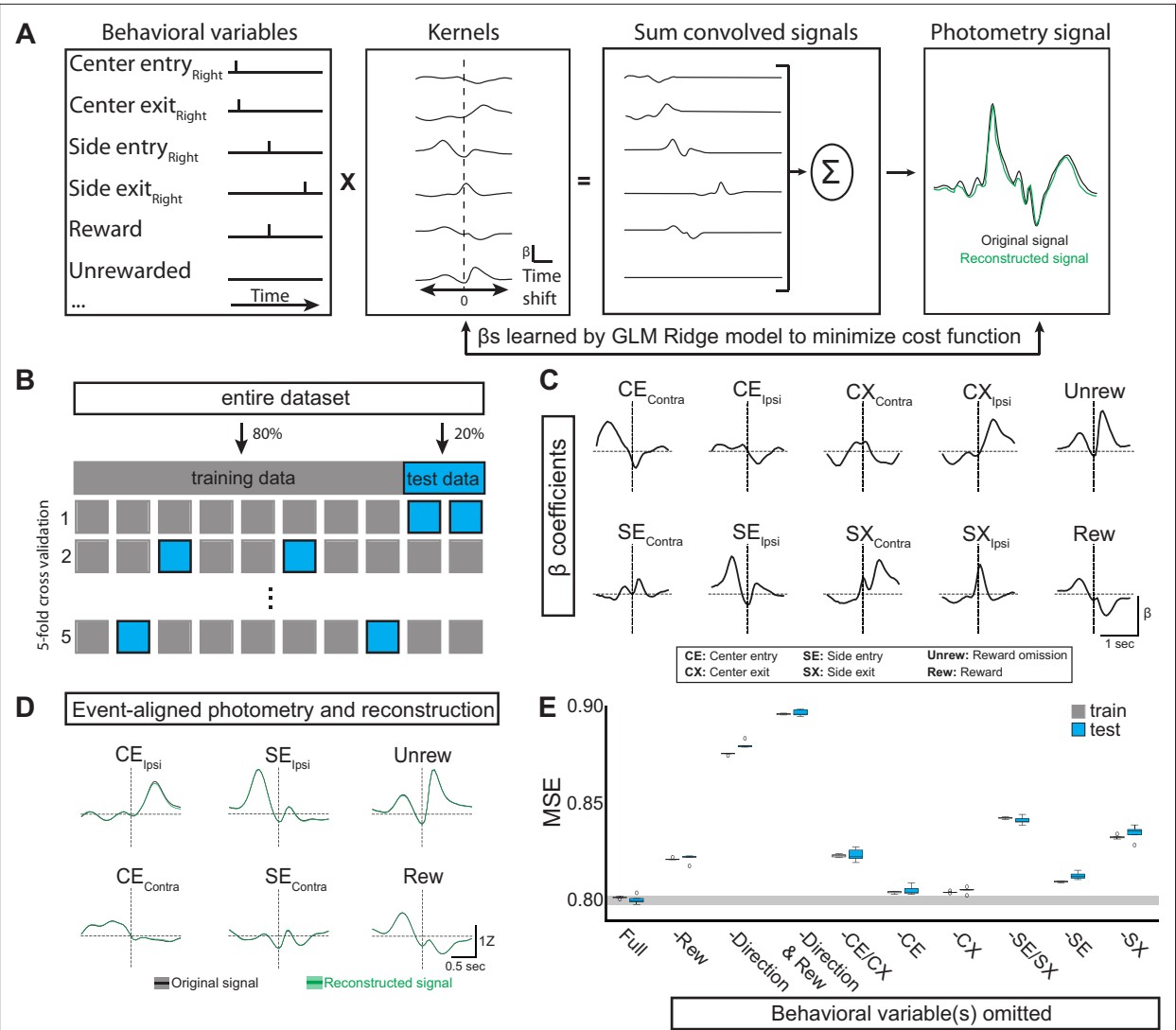

**Figure 3.** Generalized linear model of EP$^{Sst}$ neural activity during behavior. (**A**) GLM workflow: behavioral variables are convolved with their kernels. Each time shift in the kernel consists of an independent β coefficient fit jointly by minimizing a cost function. The convolved signals are then summed to generate a reconstructed signal (green) which can be directly compared to the original photometry recording (black). (**B**) The original dataset is divided into training and test datasets. The GLM is fit on the training data and evaluated on the test data using mean squared error (MSE). Following a grid search that compared multiple regularization types (ridge, elastic net, ordinary least squared) in combination with a large hyperparameter space, ridge regression (α=1) was found to give the smallest error following cross-validation. (**C**) Kernels for the ten behavioral variables included as features in the GLM (full model). Behavioral predictors gave information regarding choice (Ipsi/Contra), reward, and port entry and exit. (**D**) Average original (black) and reconstructed (green, using full model) photometry signals across trials aligned to behavioral events (solid line = mean, shaded area = SEM, R²=0.19 SD = 0.001, n=male 6 mice). (**E**) Box plots showing MSE for the full model (leftmost plot labeled 'Full') and models in which selected behavioral predictor(s) were omitted (See materials and methods). MSE for both the train (gray) and test (blue) datasets for each model are shown (Boxes represent the three quartiles (25%, 50%, and 75%) of the data and whiskers are 1.5*IQR, outliers are shown as dots, each model-run uses a different combination of data used for train/test split as illustrated in **B**).

The online version of this article includes the following figure supplement(s) for figure 3:

**Figure supplement 1.** Generalized linear model (GLM) reconstructions of photometry signal with a partial model.

**Figure supplement 2.** Generalized linear model (GLM) reconstructions of photometry signal with choice and reward variables omitted.

specific features of the simultaneously recorded neural signal (*Figure 1c*). Generalized linear models (GLMs) can be used to quantitatively determine which behavioral events explain the observed neural signal (*Chantranupong et al., 2023*; *Park et al., 2014*; *Engelhard et al., 2019*; *Driscoll et al., 2017*). We defined a set of behavioral variables (such as the timing of rewards, port entries/exits, etc.) as

predictors for a GLM to fit the neural data (*Figure 3a*). For each behavioral variable the GLM assigns a kernel of time-shifted beta (β) coefficients that represent the contribution of that variable to the neural signal (GCaMP6f fluorescence; *Figure 3a and c*). These kernels can then be convolved with the actual timing of behavior events in a trial and summed to create a 'reconstructed' GCaMP6f signal which is compared to the actual (original) signal (*Figure 3a*, right).

To estimate the beta coefficients, the original dataset (90/10 reward probabilities, n=6 mice, 5 sessions/mouse) is divided into training (80%) and test (20%) datasets. We fit the model on the training data and evaluated it on the test data using mean squared error (MSE, which is the cost function minimized by the model), calculated by comparing the reconstructed neural signal and the actual (original) photometry signal (*Figure 3a–b*). We tested and compared multiple regularization methods (ridge, elastic net, ordinary least squares) across a large hyperparameter space and found that ridge regression performed most consistently when evaluated with MSE (MSE = 0.80 SD = 0.001, $R^2$=0.19 SD = 0.001, *Figure 3d–e*).

We then determined which behavioral variables contributed to GLM performance by omitting variables and examining the importance of each variable to the model performance as measured by a change in MSE (*Figure 3e*, also see materials and methods). We found that omitting reward variables (both Unrew. and Rew. (*Figure 3c*)) decreased the performance of the GLM (increased MSE), indicating that the neural signal cannot be entirely explained by the choices and port entries/exits of the animal during a trial (*Figure 2e*, '- Rew'). As we observed large differences in the neural signal during ipsilateral and contralateral movements (*Figure 2c*), we tested the requirement for direction on GLM performance by collapsing the ipsilateral and contralateral port entries into a single variable void of directionality but preserving event timing (e.g. $SE_{Contra}$ and $SE_{Ipsi}$ were combined and represented as a single 'SE' variable, and repeated for CE, CX, and SX). This resulted in a large drop in GLM performance (increased MSE) indicating that the direction of the side port choice (ipsi vs contra) was critical for accurate reconstruction of the neural signal (*Figure 3e*, '- Direction' and *Figure 3—figure supplements 1 and 2*). Omitting center port entry/exit together (-CE/CX, *Figure 3e*) or individually also resulted in decreased GLM performance, but to a smaller degree than the omission of direction (*Figure 3e*). The same pattern was true for side port entry/exit (*Figure 3e*, -SE/SX'). Together, testing the GLM performance revealed that both direction and reward were important for optimal model performance supporting an interpretation that EP$^{Sst+}$ neurons signal both movement direction (ipsi vs contra) and reward aspects of a trial.

## EP$^{Sst+}$ neurons are not required for continued performance on a probabilistic switching task

EP$^{Sst+}$ neurons directly and exclusively project to the LHb a region principally implicated in evaluating negative outcomes of an action (*Wallace et al., 2017*; *Proulx et al., 2014*; *Wang et al., 2017*). Photometry recordings from EP$^{Sst+}$ neurons during behavior suggested that these neurons were actively engaged during both the action selection (ipsi vs contra side port) and outcome evaluation periods of a trial (*Figures 2 and 3*). We hypothesized that ablation of synaptic release from these neurons, thus blocking their ability to communicate with the LHb, would strongly impact the outcome evaluation phase of the task. We trained mice on the probabilistic switching task (90/10 reward probability) to reach predefined criteria where task performance was consistent over a week (See Methods, and *Figure 4—figure supplement 1a–c*). *Sst-Cre* mice were then injected with AAVs containing either Cre-dependent GFP (GFP, green, control) or Cre-dependent tetanus toxin light-chain which blocks synaptic vesicle fusion (*Zhang et al., 2015*) (Tettx, red; *Figure 4a*). Mice continued daily sessions on the task for 3 wk to allow for viral expression. Control animals showed no significant differences in behavioral performance after surgery, indicating that the surgery was well tolerated and resulted in no observable detrimental side effects barring a small increase in body weight in Tettx animals (*Figure 4—figure supplements 1 and 2*). We quantified the number of Tettx-expressing cells in the EP at the termination of behavioral testing as a percentage of the entire Sst+ population based on stereological estimates (*Miyamoto and Fukuda, 2015*; *Miyamoto and Fukuda, 2021*). We found that our injections targeted 70 ± 15.1% (mean ± SD) of the EP$^{Sst+}$ population (1070 ±230 neurons/animal, n=6 mice). In separate animals, we functionally confirmed that 3 weeks of Tettx expression in EP$^{Sst+}$ neurons were sufficient to block both optogenetically evoked IPSCs and EPSCs from EP$^{Sst+}$ axons to LHb neurons (*Figure 4—figure supplement 1j–k*).

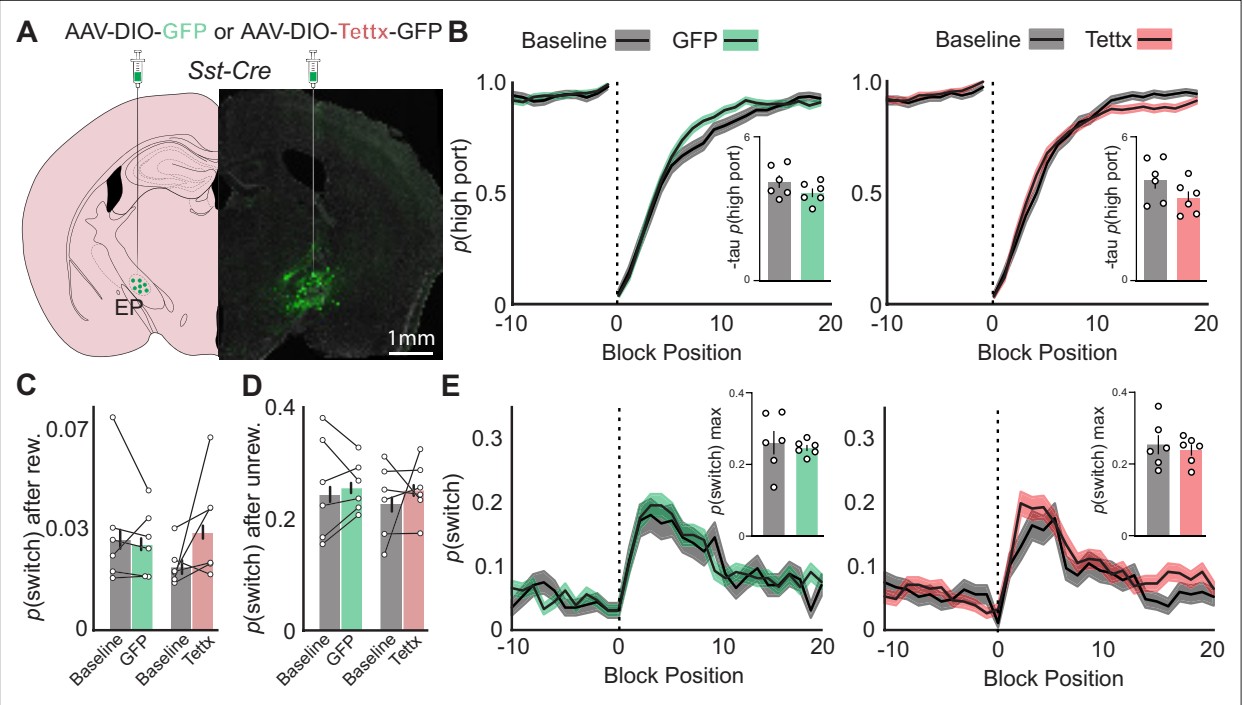

**Figure 4.** Effects of permanent genetic silencing of synaptic release from EP$^{Sst+}$ neurons on continued performance of a two-port choice probabilistic switching task. (**A**) Viral injection location resulting in Cre-dependent expression of GFP (control) or tetanus toxin in EP$^{Sst+}$ neurons (green = Tettx GFP, gray = DAPI). (**B**) The probability of choosing the highly rewarded port (*p* high port) around a block transition (dotted vertical line) for GFP (control, left) or Tettx (right) injected mice (gray = 5 d prior to adeno-associated virus (AAV) injection, green/red = days 21–30 post-injection; black line = mean, shaded area = SEM). (Insets) Bar plot showing tau$_{p(high\ port)}$ (time constant) calculated from an exponential fit to the first 20 trials following a block transition for each animal (circles) (bar = mean, error bar = SEM) before and after AAV injection. (**C**) *p*(switch) for trials following a rewarded trial for GFP (green) and Tettx (red) injected animals (bar = mean, error bar = 95% CI). (**D**) *p*(switch) for trials following an unrewarded trial for GFP (green) and Tettx (red) injected animals (bar = mean, error bar = 95% CI). (**E**) The probability of choosing different side ports on consecutive trials (*p* switch) around a block transition (dotted vertical line) for GFP (control, left) or Tettx (right) injected mice (gray = 5 d prior to AAV injection, green/red = days 21–30 post-injection; black line = mean, shaded area = SEM). (Insets) Bar plot showing the maximum *p*(switch) in the 20 trials that follow a block transition for each animal (circles) (bar = mean, error bar = SEM) before and after AAV injection. For (**B–E**) n=6 male GFP control and n=6 male Tettx mice, five sessions/mouse before AAV inj. and 10 sessions/mouse after AAV injection, GFP control = 15,120 trials before, 34,523 trials after; Tettx = 17,528 trials before, 32,761 trials after.

The online version of this article includes the following figure supplement(s) for figure 4:

**Figure supplement 1.** Additional behavioral performance metrics before and after viral injection and electrophysiological validation of Tettx effects on GABA/glutamate cotransmission from EP$^{Sst+}$ neurons.

**Figure supplement 2.** Body weight and total number of trials per session before and after viral injection.

**Figure supplement 3.** Psychometric curves of port choice before and after adeno-associated virus (AAV) injection.

We then compared behavioral performance on the task before AAV injection to sessions collected 3 week post-injection in both control and Tettx groups (*Figure 4*). Both groups performed well before and after viral injection, selecting the high reward port around a block transition similarly with no significant differences between groups (*Figure 4b*). Control and Tettx groups also showed no significant change in *p*(switch) around a block transition or following rewarded and unrewarded trials (*Figure 4c–e*) indicating that the sensitivity of the animal to detect the outcome of the previous trial and respond on subsequent trials was not significantly perturbed. All other behavioral metrics (ITI duration, trial duration, *p*(reward), port bias, etc.) were unchanged between groups or when compared before and after AAV injection (*Figure 4—figure supplement 1d–i*). Consistent with our other measures the mouse behavioral strategy as assessed by the RFLR model was also unperturbed between groups (*Figure 4—figure supplements 1m and 3a-b*). Together these data indicate that ablation of both GABA and glutamate release from EP$^{Sst+}$ neurons is not sufficient to result in profound behavioral performance changes in animals well-trained on the probabilistic switching task despite strong modulation of EP$^{Sst+}$ activity during a trial as reported by fiber photometry (*Figure 2*).

# Genetic deletion of synaptic glutamate release from EP$^{Sst+}$ neurons during the probabilistic switching task

EP$^{Sst+}$ neurons simultaneously cotransmit both GABA and glutamate onto individual neurons in the LHb (*Wallace et al., 2017*; *Kim et al., 2022*; *Shabel et al., 2014*). Although studies have suggested that the primary effect of EP$^{Sst+}$ cotransmission in LHb is excitatory in vitro (*Shabel et al., 2014*), the in vivo effects of EP$^{Sst+}$ neurons on LHb are unknown. Additionally, the ratio of GABA/glutamate cotransmitted from EP$^{Sst+}$ neurons has been shown to be plastic following exposure to environmental stressors and drugs of abuse possibly altering the net effect on LHb activity (*Shabel et al., 2014*; *Meye et al., 2016*). We reasoned that altering the ratio of GABA/glutamate cotransmission by genetic deletion of the vesicular glutamate transporter (vGluT2, *Slc17a6*) might have stronger effects on downstream LHb activity and associated behaviors than deleting both GABA and glutamate release together (*Figure 4*).

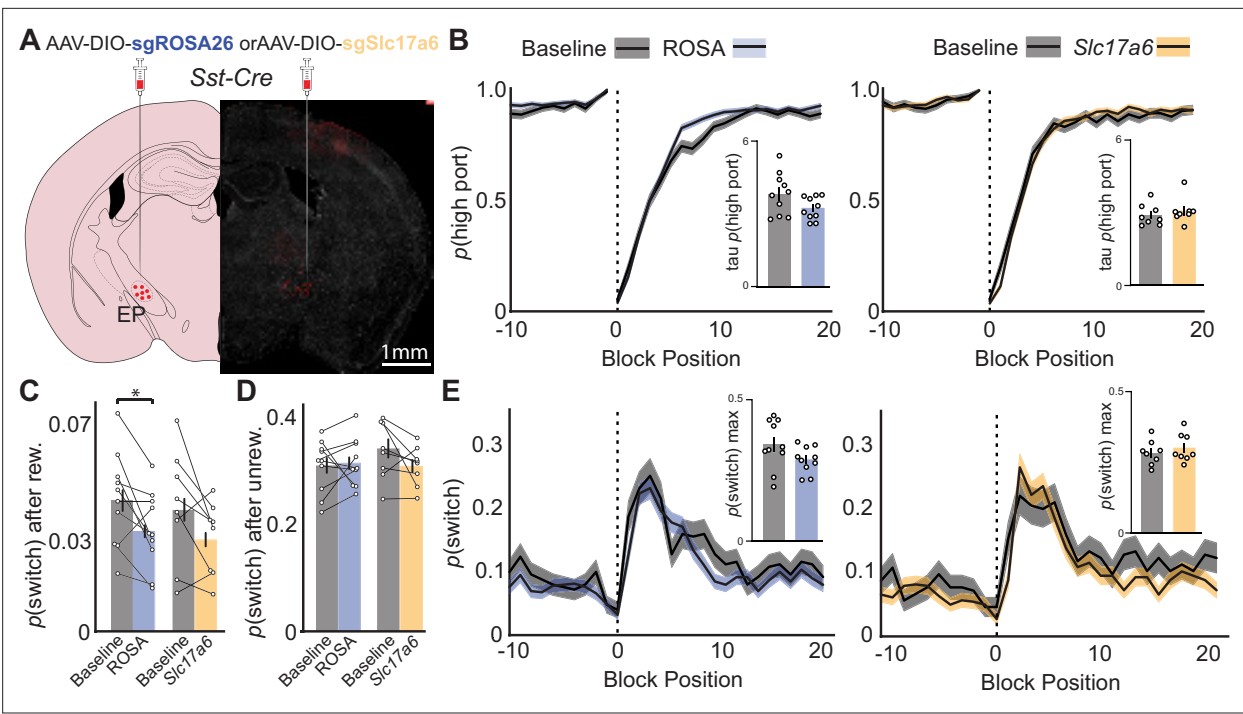

**Figure 5.** Effects of CRISPR Cas9 deletion of synaptic glutamate release from EP$^{Sst+}$ neurons on continued performance of a two-port choice probabilistic switching task. (**A**) Viral injection location resulting in Cre-dependent expression of oChief-tdTom +SaCas9-sgRNA for ROSA26 (control) or *Slc17a6* (vGluT2) in EP$^{Sst+}$ neurons (red = tdTomato, gray = DAPI). (**B**) The probability of choosing the highly rewarded port (*p* high port) around a block transition (dotted vertical line) for sgROSA (control, left) or sg*Slc17a6* (right) injected mice (gray = 5 d prior to adeno-associated virus (AAV) injection, blue/orange = days 21–30 post-injection; black line = mean, shaded area = SEM). (Insets) Bar plot showing tau$_{p(high port)}$ (time constant) calculated from an exponential fit to the first 20 trials following a block transition for each animal (circles) (bar = mean, error bar = SEM) before and after AAV injection. (**C**) *p*(switch) for trials following a rewarded trial for sgROSA26 (blue) and sg*Slc17a6* (orange) injected animals (bar = mean, error bar = 95% CI) (paired t-test DF = 9, p-value = 0.02). (**D**) *p*(switch) for trials following an unrewarded trial for sgROSA26 (blue) and sg*Slc17a6* (orange) injected animals (bar = mean, error bar = 95% CI). (**E**) The probability of choosing different side ports on consecutive trials (*p* switch) around a block transition (dotted vertical line) for sgROSA26 (control, left) or sg*Slc17a6* (right) injected mice (gray = 5 d prior to AAV injection, green/red = days 21–30 post-injection; black line = mean, shaded area = SEM). (Insets) Bar plot showing the maximum *p*(switch) in the 20 trials that follow a block transition for each animal (circles) (bar = mean, error bar = SEM) before and after AAV injection. For (B–E) n=10 (five males, five females) sgROSA26 control and n=8 (two males, six females) sg*Slc17a6* mice, five sessions/mouse before AAV inj. and 10 sessions/mouse after AAV injection, sgROSA26 control = 17,318 trials before, 39,710 trials after; sg*Slc17a6*=13,520 trials before, 29,256 trials after.

The online version of this article includes the following figure supplement(s) for figure 5:

**Figure supplement 1.** Additional behavioral performance metrics before and after viral injection and electrophysiological validation of CRISPR-SaCas9 mediated deletion of *Slc17a6* (vGluT2) on GABA/glutamate cotransmission from EP$^{Sst+}$ neurons.

**Figure supplement 2.** Total number of trials per session and animal body weight changes before and after viral injection.

**Figure supplement 3.** Psychometric curves of port choice before and after adeno-associated virus (AAV) injection.

Similar to previous experiments, we trained *Sst-Cre* mice on the probabilistic switching task (90/10 reward probability) to criteria and then injected AAVs containing *SaCas9* with either control guide RNA for the ROSA26 locus (sgROSA) or guide RNA for the *Slc17a6* the gene encoding the vesicular glutamate transporter 2 (vGluT2) to permanently disrupt gene function (*Hunker et al., 2020*; *Figure 5A*). These animals were also injected with Cre-dependent oChief for post-hoc electrophysiological examination of synaptic transmission from EP$^{Sst+}$ axons. Indeed, as confirmed after completion of behavioral experiments (5 wk post-injection), we observed near total loss of glutamatergic transmission from *Sst +* axons as measured by voltage clamp recordings in LHb at –65 mV (*Figure 5—figure supplement 1j–k*). Importantly, when LHb neurons were held at the reversal potential for AMPA receptors large optogenetically evoked IPSCs were revealed confirming that EP$^{Sst+}$ neurons still made functional synaptic contacts with the LHb, but that they were now almost entirely GABAergic (*Figure 5—figure supplement 1j–k*).

We then compared behavioral performance on the task before AAV injection to sessions collected 3 wk post-injection in both ROSA26 (control) and *Slc17a6* (vGluT2 deletion) groups (*Figure 5*, *Figure 5—figure supplement 1a-c*). Both groups performed well before and after viral injection, selecting the high reward port around a block transition similarly with no significant differences between groups (*Figure 5b*). ROSA26 and *Slc17a6* groups also showed no significant change in p(switch) around a block transition (*Figure 5e*). Examining p(switch) following different trial outcomes revealed that animals decreased their p(switch) following a rewarded trial following AAV injection, however, this effect was similar between ROSA26 and *Slc17a6* groups (*Figure 5c*). Furthermore, p(switch) following an unrewarded outcome was not altered between groups (*Figure 5d*). We also examined p(switch) following various trial histories and combinations of choices and outcomes but did not observe any differences between groups indicating that the ability of the animal to detect the outcome of the previous trial and respond on subsequent trials was not perturbed (*Figure 5—figure supplement 1l*). All other behavioral metrics (ITI duration, trial duration, p(reward), port bias, etc.) were unchanged between groups or when compared before and after AAV injection (*Figure 5—figure supplements 1m and 2a-b*). Consistent with our other measures the mouse behavioral strategy as assessed by the RFLR model was also unperturbed between groups (*Figure 5—figure supplements 1 and 2*). We conclude that permanent deletion of glutamate release from EP$^{Sst+}$ neurons effectively converts this normally cotransmitting population into a GABAergic neuronal population (*Figure 5—figure supplement 1j–k*). This, however, is not sufficient to cause detectable behavioral changes in animals that are well-trained on the probabilistic switching task (*Figure 5*).

## Discussion

Here, we described a probabilistic switching task in which mice use their history of choices and rewards to update and guide future actions (*Stephenson-Jones et al., 2016*; *Tai et al., 2012*; *Hamid et al., 2016*; *Chantranupong et al., 2023*; *Beron et al., 2022*). We show that mice can detect block transitions when reward probabilities alternate sides and respond by an increase in switching between reward (side) ports. Animals are also sensitive to the combination of reward probabilities set in a session (90/10 vs 70/30) and modify their strategy. When the probability of receiving a reward becomes more stochastic (70/30 condition) they incorporate evidence over more trials in the past to inform future actions. As rewards become more stochastic, mice also take more trials to recover stable selection of the high-reward port following a block transition, switching between ports less frequently (*Figure 1*). Using fiber photometry, we show that populations of EP$^{Sst+}$ neurons are strongly engaged during this task during both the trial choice and outcome epochs (*Figure 2*). This observation was then reinforced by a GLM that showed significant decreases in model performance when information about choice direction or choice outcome was omitted, indicating these are important predictive behavioral variables to reconstruct the photometry signal (*Figure 3*). We then tested the necessity of EP$^{Sst+}$ neurons for continued performance on the probabilistic switching task. We found that permanent genetic blockade of synaptic release from these neurons did not result in detectable changes in task performance metrics (*Figure 4*). Finally, we tested if modifying the ratio of GABA and glutamate cotransmitted by EP$^{Sst+}$ neurons had an impact on continued task performance by genetically deleting vGluT2, thereby strongly decreasing the amount of glutamate released at synapses. This manipulation did not result in significant changes in task performance across control and vGluT2 deleted groups (*Figure 5*). Together these data suggest that despite observing ongoing, task-related activity in EP$^{Sst+}$

neurons, cotransmission from these neurons to LHb was not required for continued task performance in well-trained animals.

## Probabilistic switching task and basal ganglia circuits

Dynamic, probabilistic switching tasks have been used by many groups to examine how an animal uses its past experience to make its next choice (*Tai et al., 2012*; *Hamid et al., 2016*; *Chantranupong et al., 2023*; *Beron et al., 2022*; *Samejima et al., 2005*; *Parker et al., 2016*; *Bari et al., 2019*). While we focused on EP, other studies show that distributed circuits throughout the cortex, striatum and midbrain guide animals to flexibly choose actions in pursuit of rewards on a trial-to-trial basis and have helped inform models of reinforcement learning (*Cox and Witten, 2019*). Dopamine signaling in the striatum is critical for optimal performance on this task as it can causally guide future choice, and may also underlie motivation during longer periods by integrating reward rate (*Hamid et al., 2016*). Dopamine release has also been shown to signal reward prediction error in these tasks which takes into account the reward history and expectation an animal has trial-to-trial (*Hamid et al., 2016*; *Chantranupong et al., 2023*).

Postsynaptic to dopamine release sites in striatum, SPNs are key mediators of 'action value' in these tasks. Furthermore, in a task similar to the one described here, unilateral optogenetic stimulation of D1R SPNs in the dorsal medial striatum (DMS) immediately following center entry biased future choice in the contralateral direction indicating a causative role for SPNs in guiding choice behavior in a probabilistic switching task (*Tai et al., 2012*).

Other studies have examined groups of neurons directly downstream of D1R SPNs that are suggested to be involved in evaluating action outcomes (*Hong and Hikosaka, 2008*; *Stephenson-Jones et al., 2016*; *Shabel et al., 2012*). These EP neurons receive convergent input from both striosomal (patch) and matrix D1R SPNs distributed throughout the striatum and project exclusively to the LHb, a region critical for the processing of 'negative reward' signals (*Wallace et al., 2017*; *Stephenson-Jones et al., 2016*; *Matsumoto and Hikosaka, 2007*). In head-fixed classical conditioning tasks LHb-projecting EP neurons have been shown to increase their activity to punishments or reward omission and decrease their activity to rewards (*Hong and Hikosaka, 2008*; *Stephenson-Jones et al., 2016*). Interestingly, in a probabilistic switching task optogenetic stimulation of vGluT2 + EP neurons following side port entry, but not center entry, biased future choices away from side port paired with stimulation indicating that these neurons carried an 'anti-reward' signal (*Stephenson-Jones et al., 2016*). An important consideration with the aforementioned studies is that behavioral changes resulting from phasic modulation of EP to LHb inputs rely on the combined action of *Sst+* and an additional subset of purely glutamatergic (*Pvalb+/Slc17a6+*) EP neurons as both populations and possibly others see *Lazaridis et al., 2019* are targeted in these studies (*Stephenson-Jones et al., 2016*; *Shabel et al., 2012*). In agreement with our results described here, when EP$^{Sst+}$ neurons are targeted specifically for optogenetic excitation a bias away from stimulation was not observed indicating that in naive animals activation of these neurons is not intrinsically aversive (*Lazaridis et al., 2019*). These studies and others suggest a prominent role for various basal ganglia nuclei in distinct phases of a probabilistic switching task engaging circuitry in both the action selection (choice) and action evaluation (reward) epochs of a trial. Notably, the observed effects of causal optogenetic manipulations in both striatum and EP depend on both reward history and previous choice, variables shown to be critical for animal performance (*Beron et al., 2022*; *Bolkan et al., 2022*).

## Activity patterns of EP$^{Sst+}$ neurons

Here, we show the activity patterns of genetically defined EP$^{Sst+}$ neurons during freely moving behavior. Activity of EP$^{Sst+}$ neurons is robustly modulated on a trial-by-trial basis in the probabilistic switching task by both the direction of the choice (ipsilateral vs contralateral) and the outcome (reward vs unrewarded) of a trial (*Figures 2 and 3*). In contrast to thalamic projecting EP neurons which receive striatal input exclusively from the SPNs in the matrix compartment, EP$^{Sst+}$ neurons receive input from both limbic-associated 'striosomes' (patches) and sensorimotor-associated 'matrix' subdivisions which may contribute to activity during outcome and choice epochs, respectively (*Wallace et al., 2017*; *Stephenson-Jones et al., 2016*). Notably, phasic changes in activity during ipsilateral and contralateral movements resemble those observed in substantia nigra pars reticulata (another output nucleus of the basal ganglia) during eye saccade tasks (i.e. increased/persistent activity for ipsilateral movements

and decreased for contralateral movements) (*Hikosaka et al., 2000*; *Hikosaka and Wurtz, 1985*). Our findings are also consistent with electrophysiological studies of individual LHb-projecting GPi neurons in primates that respond to both reward-related cues and sensory cues related to the direction of a target during an eye saccade task (*Hong and Hikosaka, 2008*). Reward-related responses we observe during the outcome evaluation period of the task are consistent with those reported elsewhere, with phasic excitation following unrewarded outcomes and inhibition following reward (*Hong and Hikosaka, 2008*; *Stephenson-Jones et al., 2016*). In contrast to other reports using single neuron electrophysiological recordings of LHb projecting EP neurons, we did not observe consistent bidirectional reward prediction error (RPE) coding with our photometry measurements (*Hong and Hikosaka, 2008*; *Stephenson-Jones et al., 2016*; *Figure 2—figure supplement 2*). Instead, we observed a bidirectional value signal indicating whether or not a reward occurred and a small effect of trial history on rewarded trials only (*Figure 2—figure supplement 2c–d*). RPE like responses have been observed by recording dopamine release in the striatum during a probabilistic switching task similar to the one we describe here (*Chantranupong et al., 2023*), therefore, these features may only be present in a subset of EP$^{Sst+}$ neurons, or are below our detection threshold with photometry recordings. Alternatively, LHb-projecting GPi neurons in macaques show both positive and negative reward modulations which may obscure observation of RPE signals with photometry recordings that arise from population activity of genetically defined neurons (*Hong and Hikosaka, 2008*).

Despite strong engagement of EP$^{Sst+}$ neuronal activity during the task, we were surprised to find that neither complete blockade of synaptic release or modification of the ratio of GABA/glutamate cotransmitted was sufficient to alter performance of well-trained animals on the task. Whereas single-cell sequencing has shown EP$^{Sst+}$ neurons do not express *Slc17a7/8* (vGluT1 or vGluT3) (*Wallace et al., 2017*; *Hashikawa et al., 2020*) it is possible that these genes could be upregulated following CRISPR-mediated deletion of *Slc17a6*. However, in vitro synaptic physiology demonstrates EPSCs are significant suppressed, (*Figure 5—figure supplement 1J-K*) and, therefore, can conclude it is unlikely to occur to a significant degree in our experiments. These behavioral results may point to additional parallel circuits or rapid homeostatic plasticity in LHb which compensates altered EP$^{Sst+}$ output during the gradual expression of viral constructs. Alternatively, activity in EP$^{Sst+}$ neurons (and subsequent cotransmission in LHb) may be required as the animal learns the structure of the probabilistic switching task, but then no longer required in well-trained animals that have learned the sequence of actions needed for high behavioral performance, akin to those described for motor cortex or subregions of striatum (*Kawai et al., 2015*; *Reinhold et al., 2023*). Finally, the stochastic nature of behavior in this task may require higher power for differentiating effects than available in this set of experiments. For example, ablating EP$^{Sst+}$ neurons may have effects on very small subsets of trial types that we haven't characterized due to insufficient statistical power (i.e. switch trials).

## Functions for the EP in the probabilistic switching task

A sequence of training steps is used to instruct an animal to perform the probabilistic switching task we describe here (see Methods). Once they learn the progression of a trial (i.e. poke in center to begin then poke side for reward) we introduce a 'block structure' where the high reward port switches sides following a pre-determined number of rewarded trials progressively growing to 50 reward blocks with side ports delivering rewards at 90% (high port) and 10% (low port) of trials (*Figure 1*). Prior to any photometry recording or synaptic manipulation of EP, animals must reach a predetermined ('expert') criterion and consistently perform at this level for several days (at least five consecutive sessions). There are subtle changes in performance such as decreased trial duration (see *Figure 5—figure supplement 1*) once the animals reach criterion, but by and large their performance has plateaued and stabilized. Critically, even though well-trained animals perform consistently, they do not perform the task "habitually" (i.e. they are sensitive to devaluation and will not perform if they are not thirsty, data not shown). Also, well-trained animals continue to evaluate the previous choice and trial outcome to inform future decisions, engaging in trial-and-error action updating. However, expert animals have clearly mastered the sequence of actions required to move between ports and consume rewards. Perhaps our behavioral results demonstrate that long-term manipulations of EP$^{Sst+}$ neurons do not affect continued performance on the task because this circuit is required for earlier stages of learning. Importantly, our results do not indicate that these neurons have no role in shaping initial task acquisition, particularly while the animals learn the location of rewards and the action sequence required to

acquire them. Future studies should examine the function of EP$^{Sst+}$ neurons in learning action/outcome associations prior to the crystallization of action sequences that lead to reward. Conversely, different populations of EP neurons that are not examined here, such as the thalamic projecting *Pvalb+/Slc32a1* population (*Wallace et al., 2017*), may play a critical role in executing learned action sequences as seen in studies using a forelimb lever pressing task (*Dhawale et al., 2021*).

### EP cotransmission and influence on LHb activity patterns

Major questions remain regarding how neurons in the LHb integrate and interpret signals from GABA/ glutamate cotransmitting inputs from EP$^{Sst+}$ neurons. EP$^{Sst+}$ input appears to target a subregion of lateral LHb comprising the lateral and oval subregions (*Wallace et al., 2017*; *Wallace et al., 2020*; *Andres et al., 1999*). In vitro cell-attached recordings from LHb show that most neurons respond to optogenetic stimulation of EP input with increases in spiking, however these recordings were performed in conditions where channelrhodopsin was expressed in all LHb projecting EP neurons possibly leading to a bias towards excitation (*Shabel et al., 2014*; *Meye et al., 2016*). Studies examining mPSCs or using minimal optogenetic stimulation of EP$^{Sst+}$ axons have demonstrated that individual release sites and/or synaptic vesicles can cotransmit GABA and glutamate (*Kim et al., 2022*; *Shabel et al., 2014*). Using targeted optogenetic stimulation of multiple distinct EP$^{Sst+}$ inputs onto a single LHb neuron, we found that the amplitudes of the EPSC and IPSC were correlated within a cell, but the ratio varied between cells. This indicates that when exclusively examining EP$^{Sst+}$ inputs an individual LHb neuron may be excited or inhibited depending on the ratio set by the postsynaptic receptor composition (*Kim et al., 2022*). Additional experiments need to examine how this diversity translates to an in vivo setting where the postsynaptic membrane potential is not clamped and could respond differently to cotransmission. Recent studies have demonstrated rapid, behaviorally induced plasticity in individual LHb neurons following a stressful tail-shock protocol (*Shabel et al., 2019*). Remarkably, LHb neurons change the sign of their responses (negative going to positive going) following sucrose reward delivery following stress (*Shabel et al., 2019*). It is tempting to speculate that GABA/glutamate cotransmitting synapses undergo plasticity to control LHb output by modifying the ratio of GABA/glutamate cotransmission under these or other environmental/behavioral changes (*Wallace and Sabatini, 2023*).

## Materials and methods

**Key resources table**

| Reagent type (species) or resource | Designation | Source or reference | Identifiers | Additional information |
|---|---|---|---|---|
| Strain (*Mus musculus*) | C57Bl/6 J | The Jackson Laboratory | Cat# JAX:000664 RRID:IMSR_JAX:000664 | |
| Genetic reagent (*Mus musculus*) | Sst-IRES-Cre | The Jackson Laboratory | Cat# JAX: 013044 RRID:IMSR_JAX:013044 | |
| Genetic reagent (*Mus musculus*) | Sst-IRES-Flpo | The Jackson Laboratory | Cat# JAX: 031629 RRID:IMSR_JAX:031629 | |
| Genetic reagent (*Mus musculus*) | Pvalb-2A-Flp | The Jackson Laboratory | Cat# JAX: 022730 RRID:IMSR_JAX:022730 | |
| Genetic reagent (AAV) | AAV1-Syn-FLEX-GCaMP6f | Addgene | Addgene# 100833 RRID:Addgene_100833 | |
| Genetic reagent (AAV) | AAV8-CAG-FLEX-tdTomato | Addgene | Addgene# 51503 RRID:Addgene_51503 | |
| Genetic reagent (AAV) | AAVrg-Ef1a-fDIO-Cre | Addgene | Addgene# 121675 RRID:Addgene_121675 | |
| Genetic reagent (AAV) | AAV8-Syn-FLEX-TeLC-P2A-GFP | Addgene | Addgene# 135391 RRID:Addgene_135391 | |
| Genetic reagent (AAV) | AAV8-Syn-DIO-EGFP | Addgene | Addgene# 50457 RRID:Addgene_50457 | |

*Continued on next page*

*Continued*

| Reagent type (species) or resource | Designation | Source or reference | Identifiers | Additional information |
|---|---|---|---|---|
| Genetic reagent (AAV) | AAV1-CMV-FLEX-SaCas9-sgSlc17a6 | Addgene | Addgene# 124847 RRID:Addgene_124847 | |
| Genetic reagent (AAV) | AAV1- CMV-FLEX-SaCas9-sgROSA26 | Addgene | Addgene# 159914 RRID:Addgene_159914 | |
| Genetic reagent (AAV) | AAV8-Ef1a-DIO-oChief-tdTomato | Addgene | Addgene# 51094 RRID:Addgene_51094 | |
| Antibody | anti-GFP (chicken polyclonal) | Invitrogen | Cat# A10262 RRID:AB_2534023 | IF(1:1000) |
| Antibody | anti-mcherry (rabbit polyclonal) | Abcam | Cat# Ab167453 RRID:AB_2571870 | IF(1:500) |
| Software, algorithm | MATLAB (R2015a) | MathWorks | RRID:SCR_001622 | |
| Software, algorithm | Fiji | PMID:22743772 | RRID:SCR_002285 | https://imagej.net/Fiji |
| Software, algorithm | Python | https://www.python.org/ | RRID:SCR_008394 | https://www.python.org/ |

## Mice

The following mouse strains/lines were used in this study: C57BL/6 J (The Jackson Laboratory, Stock # 000664), *Sst*-IRES-Cre (The Jackson Laboratory, Stock # 013044), *Sst*-IRES-Flpo (The Jackson Laboratory, Stock # 031629), and *Pvalb*-2A-Flp (The Jackson Laboratory, Stock # 022730). Animals were kept on a 12:12 reverse light/dark cycle under standard housing conditions. All procedures were performed in accordance with protocols approved by the Harvard Standing Committee on Animal Care or the Boston University Institutional Animal Care and Use Committee following guidelines described in the U.S. National Institutes of Health Guide for the Care and Use of Laboratory Animals.

## Adeno-associated viruses (AAVs)

Recombinant AAVs used for fiber photometry measurements (AAV1-Syn-FLEX-GCaMP6f, AAV8-CAG-FLEX-tdTomato (Addgene #100833 and #51503, respectively), and AAVrg-Ef1a-fDIO-Cre, Addgene #121675), tetanus toxin experiments (AAV8-Syn-FLEX-TeLC-P2A-GFP was a gift from Dr. Fan Wang, AAV8-Syn-DIO-EGFP (Addgene # 135391 and #50457, respectively)) and *Slc17a6* knockout experiments (AAV1-CMV-FLEX-SaCas9-sg*Slc17a6* (Addgene #124847), AAV1- CMV-FLEX-SaCas9-sgROSA26 gift from Dr Larry Zweifel and AAV8-Ef1a-DIO-oChief-tdTomato (Addgene#51094)) were commercially obtained from the Boston Children's Hospital Viral Core or directly from Addgene. Virus aliquots were stored at –80 °C, and were injected at a concentration of approximately $10^{11}$ or $10^{12}$ GC/ml.

## Stereotaxic surgeries

Adult mice were anesthetized with isoflurane (5%) and placed in a small animal stereotaxic frame (David Kopf Instruments). After exposing the skull under aseptic conditions, viruses were injected through a pulled glass pipette at a rate of 50 nL/min using a UMP3 microsyringe pump (World Precision Instruments). Pipettes were slowly withdrawn (<100 μm/s) at least 10 min after the end of the infusion. Following wound closure, mice were placed in a cage with a heating pad until their activity was recovered before returning to their home cage. Mice were given pre- and post-operative subcutaneous ketoprofen (10 mg/kg/d) or meloxicam (5 mg/kg) and buprenorphrine XR (3.25 mg/kg) as an analgesic and monitored daily for at least 4 d post-surgery. For fiber photometry experiments 200 μm diameter fibers (0.37NA Doric Lenses) with a stainless-steel ferrule were implanted ~200 μm above the injection site following the injection and adhered to the skull with cyanoacrylate glue and dental cement (C&B Metabond). Injection coordinates from Bregma for EP were –1.1 mm A/P, 2.1 mm M/L, and 4.2 mm D/V and for LHb were –1.55 mm A/P, 0.5 mm M/L, and –2.85 mm D/V. Injection volumes for specific anatomical regions and virus types were as follows EP: 250 nL (mix of GCaMP6f and

tdTom.), 200 nL (TeLC or GFP), 400 nL (1:1 mix of SaCas9-sgRNA and oChief-tdTom), or (1:1 mix of TeLC and oChief-tdTom), LHb: 200 nL (fDIO-Cre).

## Immunohistochemistry

Mice were deeply anesthetized with isoflurane and perfused transcardially with 4% paraformaldehyde in 0.1 M sodium phosphate buffer. Brains were post-fixed overnight, sunk in 30% (wt/vol) sucrose in phosphate buffered saline (PBS) and sectioned (50 µm) coronally (Freezing Microtome, Leica). Free-floating sections were permeabilized/blocked with 5% normal goat serum in PBS with 0.2% Triton X-100 (PBST) for 1 hr at room temperature and incubated with primary antibodies at 4 °C overnight and with secondary antibodies for 1 hr at room temperature in PBST supplemented with 5% normal goat serum. Brain sections were mounted on superfrost slides, dried and coverslipped with ProLong antifade reagent containing DAPI (Molecular Probes). Primary antibodies used include: chicken anti-GFP (1:1000, A10262 Invitrogen) and rabbit anti-mCherry (1:500, Ab167453 Abcam). Alexa Fluor 594- and 488-conjugated secondary antibodies to chicken and rabbit (Invitrogen) were diluted 1:500. Whole sections were imaged with an Olympus VS120/200 slide scanning microscope. Occasionally, images were linearly adjusted for brightness and contrast using ImageJ software. All images to be quantitatively compared underwent identical manipulations.

## Behavior apparatus, training, and task

https://github.com/HMS-RIC/TwoArmedBandit (*HMS-RIC, 2022*) and https://edspace.american.edu/openbehavior/project/2abt/.

The apparatus used for the behavior is as described previously (*Chantranupong et al., 2023*; *Beron et al., 2022*) with the following modifications. Clear acrylic barriers 5.5 cm in length were installed in between the center and side ports to extend the trial time to aid in better behaviorally resolved photometry recordings (these were not in place for other behavior experiments *Figures 4 and 5*). Water was delivered in ~3 µL increments. Hardware and software to control the behavior box is available online:

Mice were water restricted 1.2 ml per day prior to training, maintained at >80% initial body weight, and singly housed for the full duration of training and photometry. All training sessions were conducted in the dark under red light conditions. During the task a blue LED above the center port signals to the mouse to initiate a trial by poking in the center port. Blue LEDs above the side ports are then activated, signaling the mouse to poke in the left or right side port within 8 s. Side port reward probabilities are defined by custom software (MATLAB) and ranged from 10–90% depending on the experiment. Withdrawal from the side port ends the trial and begins a 1 s intertrial interval (ITI). An expert mouse can perform 300–700 trials in a 40 min session.

To train the mice to proficiency, they were subjected to incremental training stages. Each training session lasts for ~40 min, adjusted according to the mouse's performance. Mice progress to the next stage once they were able to complete at least 100 successful trials with at least a 75% reward rate. On the first day, they were habituated to the behavior box, with water being delivered from both side ports and triggered only by a side port poke. In the next stage, mice learned the trial structure – only a poke in center port followed by a side port poke delivers water. Then, the mice transitioned to learning the block structure, in which 50 rewarded trials on one side port triggers the reward probabilities to switch (block transition) we began probabilistic reward delivery at this stage ($p_{High}$ = 90%, $p_{Low}$ = 10%). For photometry experiments, mice performed trials in the presence of barriers in between the center and side ports. A series of transparent barriers of increasing size (small (3 cm), medium (4 cm), and long (5.5 cm)) aided in learning. Finally, the mice were subjected to fiber implantation. Following fiber implant surgeries, mice were retrained to achieve the same pre-surgery performance level. Recordings were performed 4 wk after surgery to allow for stable viral expression levels as well as a consistent and proficient level of task performance from the mice.

For experiments where we manipulated synaptic release in EP$^{Sst+}$ neurons (*Figures 4–5*) we trained mice (reward probabilities 90/10, no transparent barrier present) to the following criteria for the 5 d prior to virus injection: (**1**) $p$(highport) per session was greater than or equal to 0.80 with a variance less than 0.003, (**2**) $p$(switch) per session was less than or equal to 0.15 with a variance less than 0.001, (**3**) the $p$(left port) was between 0.45–0.55 with a variance less than 0.005, and (**4**) the animal performed at least 200 trials in a session. The mean and variance for these measurements was

calculated across the five sessions immediately preceding surgery. The criterion were determined by comparing performance profiles in separate animals and chosen based on when animals first showed stable and plateaued behavioral performance. Following surgery, mice were allowed to recover for 3 d and then continued to train for 3 wk during viral expression. Data collected during the 5 d pre-surgery period was then compared to data collected for 10 sessions following the 3 wk allotted for viral expression (i.e. days 22–31 post-surgery).

## Behavioral analysis and modeling (Recursively formulated logistic rRegression (RFLR))

Several behavioral metrics were used to characterize performance in this task and evaluate the predictive model (RFLR) used to capture these behavioral patterns. We examined the trial-to-trial dynamics around a block transition using (**1**) the probability of choosing the highly rewarded port (p(high port)) and (**2**) the probability of choosing two different ports on subsequent trials (p(switch)) as a function of trial position within a block. To quantify differences across mice in p(switch) and the time course of p(high port) following a block transition, we used single value metrics of p(switch) max and the time constant of p(high port), $\text{tau}_{\text{p(high port)}}$, respectively. Smaller $\text{tau}_{\text{p(high port)}}$ indicates a more rapid (i.e. fewer trials) recovery of stable selection of the new highly rewarded port, and a larger p(switch) max. indicates greater sensitivity of the behavior to the block transition.

The behavior was also modeled with the purpose of systematically characterizing normal and perturbed patterns of behavior across treatment groups. The above behavioral features are well captured by a RFLR (*Beron et al., 2022*), which requires three interpretable parameters to recapitulate mouse behavior. Given successful predictive accuracy across experimental conditions, we can inspect how the model captures changes in mouse behavior that result from neural perturbations. The RFLR predicts future choice via a weighted combination of choice history bias (i.e. perseveration, α), and a latent representation of evidence that gets updated by new action-outcome information on every trial (β) and decays across trials ($\tau$). Maximum likelihood parameter estimates were found using the stochastic gradient descent optimization algorithm. Fits for α, β, $\tau$ were presented for each of the experimental groups. Given comparable performance of the model across experimental conditions, comparison of parameter fits provides a method of evaluating consistency in the structure of the behavioral strategy, as defined by three parameters: a relative influence of choice perseveration, current evidence, and previous evidence (i.e. history). Psychometric curves were generated by comparing the probability that an animal chooses the left port (*p* left port chosen) on a given trial to the log-odds ($\Psi_{t+1}$) of the animal's next choice calculated from the RFLR model. Log-odds calculations were grouped into 13 bins and different conditions were fit with a logistic function of variable slope using least-squares-regression. All additional details regarding RFLR runs are available in Jupyter Notebooks online at:https://github.com/celiaberon/2ABT_behavior_models (*Beron, 2022*).

### Fiber photometry

Fiber implants on the mice were connected to a 0.37 NA patchcord (Doric Lenses, MFP_200/220/900–0.37_2 m_FCM-MF1.25, low autofluorescence epoxy), attached to a filter cube (FMC5_E1(465–480)_F1(500–540) _E2(555–570)_F2(580–680)_S, Doric Lenses). Excitation light from LEDs (Thorlabs) and was amplitude modulated at 167 Hz (470 nm excitation light, M470F3, Thorlabs; LED driver LEDD1B, Thorlabs) and 223 Hz (565 nm excitation light, M565F3, Thorlabs, LED driver LEDD1B, Thorlabs). The following excitation light power measured at the end of the patch cord were used: 470 nm = 50 µW, 565 nm = 20 µW. Signals from the photodetectors were amplified in DC mode with Newport photodetectors (NPM_2151_FOA_FC) and received by a Labjack (T7) DAC streaming at 2000 samples/s. The DAC also received synchronous information about behavior events logged from the Arduino which controls the behavior box. The following events were recorded: center port entry and exit, side port entry and exit, lick onset and offset, and LED light onset and offset.

### Photometry analysis

The frequency-modulated signals were detrended using a rolling Z-score with a time window of 1 min (12,000 samples). As the ligand-dependent changes in fluorescence measured in vivo are small (few %) and the frequency modulation is large (~100%), the variance in the frequency modulated signal is largely ligand independent. In addition, the trial structure is rapid with an average inter-trial interval

of <3 s. Thus, Z-scoring on a large time window eliminates photobleaching without affecting signal. Detrended, frequency modulated signals were frequency demodulated by calculating a spectrogram with 1 Hz steps centered on the signal carrier frequency using the MATLAB 'spectrogram' function. The spectrogram was calculated in windows of 216 samples with 108 sample overlap, corresponding to a final sampling period of 54 ms. The demodulated signal was calculated as the power averaged across an 8 Hz frequency band centered on the carrier frequency. No additional low-pass filtering was used beyond that introduced by the spectrogram windowing. For quantification of fluorescence transients as Z-scores, the demodulated signal was passed through an additional rolling Z-score (1 min window). To synchronize photometry recordings with behavior data, center port entry timestamps from the Arduino were aligned with the digital data stream indicating times of center-port entries. Based on this alignment, all other port timings were aligned and used to calculate the trial-type averaged data shown in all figures. The Z-scored fluorescence signals were averaged across trials, sessions, and mice with no additional data normalization. Statistical comparisons were made by measuring the mean z-scored fluorescence signal across a 500 ms window immediately following a given behavioral event (CE, SE, SX, …) for all trials per mouse (n=6 mice).

## Generalized linear model

Photometry recordings and behavioral data used for the GLM analysis (*Figure 3*) were collected from *Sst-Flp* mice as indicated with six sessions per mouse and ~500 trials/session. These data were aligned to behavioral events to create a predictive matrix *X* (of dimensions *N x F*) and a response vector, **y** (of dimension *N*), where *N* is the number of samples recorded in a session and *F* is the number of behavioral 'predictors' in the analysis. The predictors consisted of values 0 and 1 to indicate if a behavioral event (for example a center port entry) occurred in the time bin.

For each predictive matrix, a design matrix $\varphi(X)$ (of dimensions N×*F* (2*T*+1)) was constructed from *T* time shifts forward and backward (*T*=20, 54ms each) for each feature, allowing the GLM to fit coefficients that corresponded to time-based kernels for each of the predictive features in *X*. Data from the ITI period, in which there are no task-relevant behavioral events, were excluded, and only data spanning shortly before center entry and after side-port exit were modeled. When initial and final time shifts spanned the boundary between two trials, the overlapped data were included twice (once in each of the trials on either side of the boundary) to ensure sufficient representation of each event in training and test datasets.

To evaluate the performance of the GLMs and determine which model and hyperparameter set was best, we performed a grid search across elastic net, ordinary least squares, and ridge regressions. For each model run, a 10-fold group shuffle split (GSS) by trial was applied to the training set to obtain cross-validated ranges for the MSEs, based on an 80–20 training/test split within each of the 10 GSS folds. Ridge regression ($\alpha$=1) was determined to be the best model based on the lowest and least variable MSE score (0.80, SD = 0.001). We then tested the effect of omitting behavioral variables on the GLM performance (*Figure 3e*) and re-fit the GLM with fivefold GSS to obtain cross-validated ranges for the MSE values used in the box plots. All ten variables used to train the 'Full' GLM are listed in *Figure 3c*. In *Figure 3e* variable(s) were omitted to test how they contributed to GLM performance. Omitted variables are defined as follows: **_-Rew_**=Rew + Unrew removed, **_-Direction_**=Ipsi/Contra info removed from CE, CX, SE, SX, **_-Direction &_** Rew = Ipsi/Contra info removed from all variables + Rew/Unrew removed, **_-CE/CX_** =Ipsi/Contra CE and CX removed, **_-CE_**=Ipsi/contra CE removed, **_-CX_**=Ipsi/contra CX removed, **_-SE/SX_** = Ipsi/Contra SE and SX removed, **_-SE_**=Ipsi/contra SE removed, **_-SX_**=Ipsi/contra SX removed. For the model chosen (Ridge Regression), the $J(X,y) = \|y - X\beta\|_2^2 + \alpha \|\beta\|_2^2$ algorithm minimizes an associated cost function with respect to the fitted coefficients as follows, where *J* is the cost function to be minimized, *X* is the design matrix (set of time-shifted behavioral events), *y* is the response vector (GCaMP6f), $\beta$ is the set of fitted coefficients, $\|a\|^2_2$ is the sum of the squared entries in vector a, and $\alpha$ is the regularization parameter.

All additional details regarding GLM runs are available in Jupyter Notebooks online at: https://github.com/mwall2017/sabatini-glm-workflow (copy archieved *Wallace, 2025*).

## Acute brain slice preparation

Brain slices were obtained from 50 to 150-d-old mice (both male and female) using standard techniques. Mice were anesthetized by isoflurane inhalation and perfused transcardially with ice-cold

artificial cerebrospinal fluid (ACSF) containing (in mM) 125 NaCl, 2.5 KCl, 25 NaHCO$_3$, 2 CaCl$_2$, 1 MgCl$_2$, 1.25 NaH$_2$PO$_4$ and 25 glucose (295 mOsm/kg). Cerebral hemispheres were removed, blocked, and transferred into a slicing chamber containing ice-cold ACSF. Coronal slices of LHb (250 µm thick) were cut with a Leica VT1000s/VT1200s vibratome in ice-cold ACSF, transferred for 10 min to a holding chamber containing choline-based solution (consisting of (in mM): 110 choline chloride, 25 NaHCO$_3$, 2.5 KCl, 7 MgCl$_2$, 0.5 CaCl$_2$, 1.25 NaH$_2$PO$_4$, 25 glucose, 11.6 ascorbic acid, and 3.1 pyruvic acid) at 34 °C then transferred to a secondary holding chamber containing ACSF at 34 °C for 10 min and subsequently maintained at room temperature (20–22°C) until use. All recordings were obtained within 4 hr of slicing. Both choline solution and ACSF were constantly bubbled with 95% O2/5% CO2.

## Electrophysiology

Individual slices were transferred to a recording chamber mounted on an upright microscope and continuously superfused (4 ml/min) with room temperature ACSF. Cells were visualized through a 60 X or 40 X water immersion objective with infrared differential interference and epifluorescence to identify regions displaying the highest density of ChR2 + axons. Epifluorescence was attenuated and used sparingly to minimize ChR2 activation prior to recording. Patch pipettes (2–4 MΩ) pulled from borosilicate glass (Sutter Instruments) were filled with an internal solution containing (in mM) 135 CsMeSO$_3$, 10 HEPES, 1 EGTA, 3.3 QX-314 (Cl– salt), 4 Mg-ATP, 0.3 Na-GTP, 8 Na2-Phosphocreatine (pH 7.3 adjusted with CsOH; 295 mOsm/kg) for voltage-clamp recordings. Membrane currents were amplified and low-pass filtered at 3 kHz using a Multiclamp 700B amplifier (Molecular Devices, Sunnyvale, CA), digitized at 10 kHz and acquired using National Instruments acquisition boards and a custom version of ScanImage (*Pologruto et al., 2003*) (available upon request or from https://openwiki.janelia.org/wiki/display/ephus/ScanImage) written in MATLAB (Mathworks, Natick, MA) or PClamp 11 (Molecular Devices). Electrophysiology data were analyzed offline in MATLAB and Clampfit. The approximate location of the recorded neuron was confirmed after termination of the recording using a 4 X objective to visualize the pipette tip, while referencing an anatomical atlas (Allen Institute Reference Atlas). For analyses in *Figure 4—figure supplement 1* and *Figure 5—figure supplement 1*, the peak amplitude of PSCs measured were averaged across at least 10 trials. To activate oChief-expressing cells and axons, light from a 473 nm laser (Optoengine) was focused on the back aperture of the microscope objective to produce wide-field illumination of the recorded cell. For voltage clamp experiments, brief pulses of light (1ms duration; 10 mW·mm$^{-2}$ under the objective) were delivered at the recording site at 20 s intervals under control of the acquisition software.

## Statistics and design

Experiments in *Figure 4*, *Figure 5* were conducted blind to virus injected and subjects assigned to each group were randomized. If an animal did not reach our performance criteria in the baseline period (described above) it was excluded from the experiment. In *Figure 1*, *Figure 1—figure supplements 1 and 2* we used one-way ANOVA with Tukey's post hoc test for multiple comparisons (p-values are designated as: *p<0.05, **p<0.01, ***p<0.001). In *Figure 2*, *Figure 2—figure supplements 1 and 2*, we used paired t-tests for two groups or repeated measures ANOVA with Tukey's post hoc test for multiple comparisons for three groups. In and *Figure 4*, *Figure 4—figure supplements 1–3* and *Figure 5*, *Figure 5—figure supplements 1–3* we used a two-way ANOVA with a Sidak's post hoc for multiple comparisons, when comparing before and after AAV injection and between control and Tettx groups. Mann Whitney test was used for comparisons of EPSC/IPSC amplitudes in *Figure 4—figure supplement 1*, *Figure 5—figure supplement 1*. For psychometric curves in *Figure 1—figure supplement 2h* we used an Extra sum-of-squares F-test to ask if the groups were better fit by one curve or individual curves to test if the groups differed from each other.

## Acknowledgements

The authors thank Julia Williams for assistance in behavioral training, James Levasseur for animal husbandry and genotyping, and Lillian Worth for administrative assistance. We thank Shay Neufeld for initial task development, box design, and behavioral analysis. We thank Jeffrey Markowitz for assistance in developing the fiber photometry system and the members of the Sabatini and Wallace labs for helpful discussions and advice. The HMS Research Instrumentation Core (Ofer Mazor and Pavel Gorelik) were essential in the development of the behavioral boxes, PCB fabrication, and design

of Ardunio/Matlab code. This work was supported by the Brain Behavior Research Foundation, the Whitehall foundation, NINDS R00NS105883, and NIMH R01MH133608, MLW as well as the Howard Hughes Medical Institute and NINDS R01NS103226, BLS.

## Additional information

### Funding

| Funder | Grant reference number | Author |
|---|---|---|
| Brain and Behavior Research Foundation | | Michael Wallace |
| Whitehall Foundation | | Yijun Liu<br>Jesse White<br>Emily Kraft<br>Michael Wallace |
| National Institute of Neurological Disorders and Stroke | R00NS105883 | Julianna Locantore<br>Michael Wallace |
| National Institute of Mental Health | R01MH133608 | Emily Kraft<br>Michael Wallace |
| National Institute of Neurological Disorders and Stroke | R01NS103226 | Bernardo Sabatini |
| Howard Hughes Medical Institute | | Bernardo Sabatini |

The funders had no role in study design, data collection and interpretation, or the decision to submit the work for publication.

### Author contributions

Julianna Locantore, Data curation, Investigation, Methodology; Yijun Liu, Data curation, Formal analysis, Investigation, Methodology; Jesse White, Janet Berrios Wallace, Emily Kraft, Data curation, Investigation; Celia Beron, Software; Bernardo Sabatini, Conceptualization, Resources, Software, Supervision, Project administration; Michael Wallace, Conceptualization, Resources, Data curation, Formal analysis, Supervision, Investigation, Methodology, Project administration

### Author ORCIDs

Yijun Liu ⬤ https://orcid.org/0009-0000-5612-7666
Celia Beron ⬤ https://orcid.org/0000-0003-4289-253X
Bernardo Sabatini ⬤ https://orcid.org/0000-0003-0095-9177
Michael Wallace ⬤ https://orcid.org/0000-0002-7270-8521

### Ethics

All procedures were performed in accordance with protocols approved by the Harvard Standing Committee on Animal Care or the Boston University Institutional Animal Care and Use Committee following guidelines described in the U.S. National Institutes of Health Guide for the Care and Use of Laboratory Animals (HMS IACUC protocol #IS00000571; BU IACUC protocol #PROTO202100002). All surgery performed under isoflurane anesthesia.

Reviewer #1 (Public review): https://doi.org/10.7554/eLife.100488.3.sa1
Reviewer #2 (Public review): https://doi.org/10.7554/eLife.100488.3.sa2
Reviewer #3 (Public review): https://doi.org/10.7554/eLife.100488.3.sa3
Author response https://doi.org/10.7554/eLife.100488.3.sa4

## Additional files

### Supplementary files
MDAR checklist

### Data availability
All data (behavioral and photometry) are available on Harvard Dataverse (https://doi.org/10.7910/DVN/3VBKTF) and analysis code are available on GitHub.

The following dataset was generated:

| Author(s) | Year | Dataset title | Dataset URL | Database and Identifier |
|---|---|---|---|---|
| Locantore JR, Liu Y, White J, Wallace JB, Beron CC, Kraft E, Sabatini BL, Wallace ML | 2024 | Mixed representations of choice direction and outcome by GABA/glutamate cotransmitting neurons in the entopeduncular nucleus | https://doi.org/10.7910/DVN/3VBKTF | Harvard Dataverse, 10.7910/DVN/3VBKTF |

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
