## [Editor Report · eLife Assessment]

Somatostatin-expressing neurons of the entopeduncular nucleus (EPNSst+) provide a limbic output of the basal ganglia and co-release GABA and Glutamate in their projection to the lateral habenula, a structure that is key for reward-based learning. Combining fiber photometry and computational modeling, the authors provide **compelling** evidence that EPNSst+ neural activity represents movement, choice direction and reward outcomes in a probabilistic switching task but, surprisingly, neither chronic genetic silencing of these neurons nor selectively elimination glutamate release affected behavioral performance in well-trained animals. This **valuable** study shows that despite its representation of key task variables, EPNSst+ neurons are dispensable for ongoing performance in a task requiring outcome monitoring to optimize reward. This work will be of interest to those interested in neural circuits, learning, and/or decision making.

---

## [Referee Report · Reviewer #1 (Public review)]

Summary:

In this series of studies, Locantore et al. investigated the role of SST-expressing neurons in the entopeduncular nucleus (EPNSst+) in probabilistic switching tasks, a paradigm that requires continued learning to guide future actions. In prior work, this group had demonstrated EPNSst+ neurons co-release both glutamate and GABA and project to the lateral habenula (LHb), and LHb activity is also necessary for outcome evaluation necessary for performance in probabilistic decision-making tasks. Previous slice physiology works have shown that the balance of glutamate/GABA co-release is plastic, altering the net effect of EPN on downstream brain areas and neural circuit function. The authors used a combination of in vivo calcium monitoring with fiber photometry and computational modelling to demonstrate that EPNSst+ neural activity represents movement, choice direction and reward outcomes in their behavioral task. However, viral-genetic manipulations to synaptically silence these neurons or selectively eliminate glutamate release had no effect on behavioral performance in well-trained animals. The authors conclude that despite their representation of task variables, EPN Sst+ neuron synaptic output is dispensable for task performance.

Strengths and Weaknesses:

Overall, the manuscript is exceptionally scholarly, with a clear articulation of the scientific question and a discussion of the findings and their limitations. The analyses and interpretations are careful and rigorous. This review appreciates the thorough explanation of the behavioral modelling and GLM for deconvolving the photometry signal around behavioral events, and the transparency and thoroughness of the analyses in the supplemental figures. This extra care has the result of increasing the accessibility for non-experts, and bolsters confidence in the results. To bolster a reader's understanding of results, we suggest it would be interesting to see the same mouse represented across panels (i.e. Fig 1 F-J, Supp 1 F,K etc i.e via inclusion of faint hash lines connecting individual data points across variables). Additionally, Fig 3E demonstrates that eliminating the 'reward' and 'choice and reward' terms from the GLM significantly worsens model performance; to demonstrate the magnitude of this effect, it would be interesting to include a reconstruction of the photometry signal after holding out of both or one of these terms, alongside the 'original' and 'reconstructed' photometry traces in panel D. This would help give context for how the model performance degrades by exclusion of those key terms. Finally, the authors claimed calcium activity increased following ipsilateral movements. However, figure 3C clearly shows that both SXcontra and SXisi increase beta coefficients. Instead, the choice direction may be represented in these neurons, given that beta coefficients increase following CXipsi and before SEipsi, presumably when animals make executive decisions. Could the authors clarify their interpretation on this point? Also, it is not clear if there is a photometry response related to motor parameters (i.e. head direction or locomotion, licking), which could change the interpretation of the reward outcome if it is related to a motor response; could the authors show photometry signal from representative 'high licking' or 'low licking' reward trials, or from spontaneous periods of high. Vs low locomotor speeds (if the sessions are recorded) to otherwise clarify this point?

There are a few limitations with the design and timing of the synaptic manipulations that would improve the manuscript if discussed or clarified. The authors take care to validate the intersectional genetic strategies: Tetanus Toxin virus (which eliminates synaptic vesicle fusion) or CRISPR editing of Slc17a6, which prevents glutamate loading into synaptic vesicles. The magnitude of effect in the slice physiology results are striking. However, this relies on co-infection of a second AAV to express channelrhodopsin for the purposes of validation, and it is surely the case that there will not be 100% overlap between the proportion of cells infected. Alternative means of glutamate packaging (other VGluT isoforms, other transporters etc) could also compensate for the partial absence of VGluT2, which should be discussed. The authors do not perform a complimentary experiment to delete GABA release (i.e. via VGAT editing), which is understandable, given the absence of an effect with the pan-synaptic manipulation. A more significant concern is the timing of these manipulations as the authors acknowledge. The manipulations are all done in well-trained animals, who continue to perform during the length of viral expression. Moreover, after carefully showing that mice use different strategies on the 70/30 version vs the 90/10 version of the task, only performance on the 90/10 version is assessed after the manipulation. Together, the observation that EPNsst activity does not alter performance on a well learned, 90/10 switching task decreases the impact of the findings, as this population may play a larger role during task acquisition or under more dynamic task conditions. Additional experiments could be done to strengthen the current evidence, although the limitations is transparently discussed by the authors.

Finally, intersectional strategies target LHb-projecting neurons, although in the original characterization it is not entirely clear that the LHb is the only projection target of EPNsst neurons. A projection map would help clarify this point.

Overall, the authors used a pertinent experimental paradigm and common cell-specific approaches to address a major gap in the field, which is the functional role of glutamate/GABA co-release from the major basal ganglia output nucleus in action selection and evaluation. The study is carefully conducted, their analyses are thorough, and the data are often convincing and thought-provoking. However, the limitations of their synaptic manipulations with respect to the behavioral assays reduces generalizability and to some extent the impact of their findings.

Comments on the latest version:

Specifically, they have included more thorough analyses to address several concerns related to interpreting activity patterns of EPSst+ neurons. The authors clearly point out that calcium activity increased during ipsilateral movements, and the increase was statistically larger during the choice phase (Figure 2 supplement 1F-G), indicating that these neurons may represent movement and additional factors (e.g. executive decision-making). Correspondingly, we appreciate the thorough explanation of using a GLM model to determine which behavioural variables contribute to observed physiological signals and adding the example reconstructed signal with direction and reward variables omitted in Figure 3 supplements 1 and 2.

Although no new manipulation experiment is added to the manuscript, the authors respond to common critiques related to testing the behavioural effect after the manipulations in well-trained mice. The discussion related to technical limitations, possible compensatory mechanisms and alternative interpretations is thorough and overall satisfying. Based on the behaviour modeling results, the authors speculate that animals need to integrate more evidence from the past to guide choice in a more uncertain environment (70/30 version), instead of adopting a 'win-stay, lose-shift' strategy in the more deterministic 90/10 version. The authors expand the discussion, but the possibility that EPNSst+ neurons contribute to task performance in well-trained animals under uncertainty is not directly tested. Along with other alternative explanations discussed in the manuscript, we think the paper is valuable literature for future studies to understand the basal ganglia circuits in learning and decision-making.

---

## [Referee Report · Reviewer #2 (Public review)]

Summary:

This paper aimed to determine the role EP sst+ neurons play in a probabilistic switching task.

Strengths:

- The in vivo recording of the EP sst+ neurons activity in the task is one of the strongest parts of this paper. Previous work had recorded from the EP-LHb population in rodents and primates in head fixed configurations, the recordings of this population in a freely moving context is a valuable addition to these studies and has highlighted more clearly that these neurons respond both at the time of choice and outcome.

- The use of a refined intersectional technique to record specifically the EP sst+ neurons is also an important strength of the paper. This is because previous work has shown that there are two genetically different types of glutamatergic EP neurons that project to the LHb. Previous work had not distinguished between these types in their recordings so the current results showing that the bidirectional value signaling is present in the EP sst+ population is valuable.

Weaknesses:

- One of the main weaknesses of the paper is to do with how the effect of the EP sst+ neurons on the behavior was assessed.

o All the manipulations (blocking synaptic release and blocking glutamatergic transmission) are chronic and more importantly the mice are given weeks of training after the manipulation before the behavioral effect is assessed. This means that as the authors point out in their discussion the mice will have time to adjust to the behavioral manipulation and compensate for the manipulations. The results do show that mice can adapt to these chronic manipulations and that the EP sst+ are not required to perform the task. What is unclear is whether the mice have compensated for the loss of EP sst+ neurons and whether they play a role in the task under normal conditions. Acute manipulations or chronic manipulations without additional training would be needed to assess this.

o Another weakness is that the effect of the manipulations was assessed in the 90/10 contingency version of the task. Under these contingencies, mice integrate past outcomes over fewer trials to determine their choice and animals act closer to a simple win-stay-lose switch strategy. Due to this it is unclear if the EP sst+ neurons would play a role in the task when they must integrate over a larger number of conditions in the less deterministic 70/30 version of the task. Indeed it is not clear that lesioning any other regions involved in evaluation of action outcomes such as VTA dopamine neurons, that encode reward prediction errors, would have any deficit when assessed in this way. Due to this, it's not clear if the mice have adapted to solve the task without evaluating action outcomes at all and are just acting in a more deterministic lose switch manner that would not presumably involve any of the circuitry in evaluating action outcomes.

- The authors conclude that they do not see any evidence for bidirectional prediction errors. It is not possible to conclude this. First, they see a large response in the EP sst+ neurons to the omission of an expected reward. This is what would be expected of a negative reward prediction error. There are much more specific well controlled tests for this that are commonplace in head-fixed and freely moving paradigms that could be tested to probe this. The authors do look at the effect of previous trials on the response and do not see strong consistent results, but this is not a strong formal test of what would be expected of a prediction error, either a positive or negative. The other way they assess this is by looking at the size of the responses in different recording sessions with different reward contingencies. They claim that the size of the reward expectation and prediction error should scale with the different reward probabilities. If all the reward probabilities were present in the same session this should be true as lots of others have shown for RPE. Because however this data was taken from different sessions it is not expected that the responses should scale, this is because reward prediction errors have been shown to adaptively scale to cover the range of values on offer (Tobler et al., Science 2005). A better test of positive prediction error would be to give a larger than expected reward on a subset of trials. Either way there is already evidence that responses reflect a negative prediction error in their data and more specific tests would be needed to formally rule in or out prediction error coding especially as previous recordings have shown it is present in previous primate and rodent recordings.

- There are a lot of variables in the GLM that occur extremely close in time such as the entry and exit of a port. If two variables occur closely in time and are always correlated it will be difficult if not impossible for a regression model to assign weights accurately to each event. This is not a large issue, but it is misleading to have regression kernels for port entry and exits unless the authors can show these are separable due to behavioral jitter or a lack of correlation under specific conditions, which does not seem to be the case.

---

## [Referee Report · Reviewer #3 (Public review)]

Summary:

The authors find that Sst-EPN neurons, which project to the lateral habenula, encode information about response directionality (left vs right) and outcome (rewarded vs unrewarded). Surprisingly, chronic impairment of vesicular signaling in these neurons onto their LHb targets did not impair probabilistic choice behavior.

Strengths:

Strengths of the current work include extremely detailed and thorough analysis of data at all levels, not only of the physiological data, but also an uncommonly thorough analysis of behavioral response patterns.

Weaknesses:

In this revised manuscript, the authors have addressed my earlier critiques.

---

## [Author Response]

The following is the authors’ response to the original reviews.

**Public Reviews:**

**Reviewer #1 (Public Review):**
Summary:In this series of studies, Locantore et al. investigated the role of SST-expressing neurons in the entopeduncular nucleus (EPNSst+) in probabilistic switching tasks, a paradigm that requires continued learning to guide future actions. In prior work, this group had demonstrated EPNSst+ neurons co-release both glutamate and GABA and project to the lateral habenula (LHb), and LHb activity is also necessary for outcome evaluation necessary for performance in probabilistic decision-making tasks. Previous slice physiology works have shown that the balance of glutamate/GABA co-release is plastic, altering the net effect of EPN on downstream brain areas and neural circuit function. The authors used a combination of in vivo calcium monitoring with fiber photometry and computational modeling to demonstrate that EPNSst+ neural activity represents movement, choice direction, and reward outcomes in their behavioral task. However, viral-genetic manipulations to synaptically silence these neurons or selectively eliminate glutamate release had no effect on behavioral performance in well-trained animals. The authors conclude that despite their representation of task variables, EPN Sst+ neuron synaptic output is dispensable for task performance.Strengths and Weaknesses:Overall, the manuscript is exceptionally scholarly, with a clear articulation of the scientific question and a discussion of the findings and their limitations. The analyses and interpretations are careful and rigorous. This review appreciates the thorough explanation of the behavioral modeling and GLM for deconvolving the photometry signal around behavioral events, and the transparency and thoroughness of the analyses in the supplemental figures. This extra care has the result of increasing the accessibility for non-experts, and bolsters confidence in the results.(1) To bolster a reader's understanding of results, we suggest it would be interesting to see the same mouse represented across panels (i.e. Figures 1 F-J, Supplementary Figures 1 F, K, etc i.e via the inclusion of faint hash lines connecting individual data points across variables).

Thank you for the suggestion. The same mouse is now represented in Fig 1 and Fig 1—Figure Supplement 1 as a darkened circle so it can be followed across different panels. Photometry from this mouse was used as sample date in Figure 2b and Figure 2—figure supplement 1a-b.

(2) Additionally, Figure 3E demonstrates that eliminating the 'reward' and 'choice and reward' terms from the GLM significantly worsens model performance; to demonstrate the magnitude of this effect, it would be interesting to include a reconstruction of the photometry signal after holding out of both or one of these terms, alongside the 'original' and 'reconstructed' photometry traces in panel D. This would help give context for how the model performance degrades by exclusion of those key terms.

We have now added analyses and reconstructed photometry signals from GLMs excluding important predictors in Figure 3—figure supplement 1 and 2. We use the model where both “Direction and reward” were omitted as predictors for the GLM and showed photometry reconstructions aligned to behavioral events used for the full model (Figure 3—figure supplement 1) and partial model (Figure 3—figure supplement 2) to compare model performance.

(3) Finally, the authors claimed calcium activity increased following ipsilateral movements. However, Figure 3C clearly shows that both SXcontra and SXipsi increase beta coefficients. Instead, the choice direction may be represented in these neurons, given that beta coefficients increase following CXipsi and before SEipsi, presumably when animals make executive decisions. Could the authors clarify their interpretation on this point?

We observe that calcium activity increases during ipsilateral choices as the animal moves toward the ipsilateral side port (e.g. CX_ipsi_ to SE_ipsi_; Fig 2C and Fig 3C). The animal also makes other ipsiversive movements not during the “choice” phase of a trial such as when it is returning to the center port following a contralateral choice (e.g. SX_Contra_ to CE; Fig 2—figure supplement 1F and Fig 3C). We also observe an increase in calcium activity during these ipsiversive movements (e.g. SX_Contra_ to CE), but they are not as large as those observed during the choice phase (Fig 2—figure supplement 1G). Therefore, during the choice phase of a trial, activity contains signals related to ipsilateral movement and additional factors (e.g. executive decision making).

(4) Also, it is not clear if there is a photometry response related to motor parameters (i.e. head direction or locomotion, licking), which could change the interpretation of the reward outcome if it is related to a motor response; could the authors show photometry signal from representative 'high licking' or 'low licking' reward trials, or from spontaneous periods of high vs. low locomotor speeds (if the sessions are recorded) to otherwise clarify this point?

Unfortunately, neither licks nor locomotion were recorded during the behavioral sessions when photometry was recorded. In Figure 2—figure supplement 1a we now show individual trials sorted by trial duration (time elapsed between CE and SE) to illustrate the dynamics of the photometry signal on fast vs slow trials within a session.

(5) There are a few limitations with the design and timing of the synaptic manipulations that would improve the manuscript if discussed or clarified. The authors take care to validate the intersectional genetic strategies: Tetanus Toxin virus (which eliminates synaptic vesicle fusion) or CRISPR editing of Slc17a6, which prevents glutamate loading into synaptic vesicles. The magnitude of effect in the slice physiology results is striking. However, this relies on the co-infection of a second AAV to express channelrhodopsin for the purposes of validation, and it is surely the case that there will not be 100% overlap between the proportion of cells infected.

For the Tet-tox experiments in Figure 4 we estimate approximately 70±15% of EP^Sst+^ neurons expressed Tet-tox based on our histological counts and published stereological counts in EP (Miyamoto and Fukuda, 2015). It is true that channelrhodopsin expression will not overlap 100% with cells infected by the other virus, indeed our in vitro synaptic physiology shows small residual postsynaptic currents following optogenetic stimulation either from incomplete blockade of synaptic release or neurons that expressed channelrhodopsin but not Tettx (Figure 4—figure supplement 1J-K). The same is shown for CRISPR mediated deletion of Slc17a6 (Fig 5 – Fig supplement 1J-K).

(6) Alternative means of glutamate packaging (other VGluT isoforms, other transporters, etc) could also compensate for the partial absence of VGluT2, which should be discussed.

While single cell sequencing (Wallace et al, 2017) has shown EP^Sst+^ neurons do not express Slc17a7/8 (vGlut1 or vGlut3) it is possible that these genes could be upregulated following CRISPR mediated deletion of Slc17a6, however we do not see evidence of this with our in vitro synaptic physiology (EPSCs are significant suppressed, Figure 5 – Fig supplement 1J-K) and therefore can conclude it is highly unlikely to occur to a significant degree in our experiments. This is now included in the Discussion.

(7) The authors do not perform a complimentary experiment to delete GABA release (i.e. via VGAT editing), which is understandable, given the absence of an effect with the pan-synaptic manipulation. A more significant concern is the timing of these manipulations as the authors acknowledge. The manipulations are all done in well-trained animals, who continue to perform during the length of viral expression. Moreover, after carefully showing that mice use different strategies on the 70/30 version vs the 90/10 version of the task, only performance on the 90/10 version is assessed after the manipulation. Together, the observation that EPNsst activity does not alter performance on a well-learned, 90/10 switching task decreases the impact of the findings, as this population may play a larger role during task acquisition or under more dynamic task conditions. Additional experiments could be done to strengthen the current evidence, although the limitation is transparently discussed by the authors.

As mentioned above, it is possible that a requirement for EP^Sst+^ neurons could be revealed if the experiment was conducted with different parameters (either different reward probabilities, fluctuating reward probabilities within a session, or withholding additional training during viral expression). It is difficult to predict which version of the task, if any, would be most likely to reveal a requirement for EP^Sst+^ neurons based on our results. We favor testing for EP^Sst+^ function using a new behavioral paradigm that allows us to carefully examine task learning following EP manipulations in an independent study.

(8) Finally, intersectional strategies target LHb-projecting neurons, although in the original characterization, it is not entirely clear that the LHb is the only projection target of EPNsst neurons. A projection map would help clarify this point.

In a previous study we confirmed that EP^Sst+^ neurons project exclusively to the LHb using cell-type specific rabies infection and examining all reported downstream regions for axon collaterals (Wallace et al 2017, Suppl. Fig 6F-G). When EP^Sst+^ neurons were labeled we did not observe axon collaterals in known targets of EP such as ventro-antero lateral thalamus, red nucleus, parafasicular nucleus of the thalamus, or the pedunculopontine tegmental nucleus, only in the LHb. Additionally, using single cell tracing techniques, others have shown EP neurons that exclusively project to the LHb (Parent et al, 2001).

Overall, the authors used a pertinent experimental paradigm and common cell-specific approaches to address a major gap in the field, which is the functional role of glutamate/GABA co-release from the major basal ganglia output nucleus in action selection and evaluation. The study is carefully conducted, their analyses are thorough, and the data are often convincing and thought-provoking. However, the limitations of their synaptic manipulations with respect to the behavioral assays reduce generalizability and to some extent the impact of their findings.
**Reviewer #2 (Public Review):**
Summary:This paper aimed to determine the role EP sst+ neurons play in a probabilistic switching task.Strengths:The in vivo recording of the EP sst+ neuron activity in the task is one of the strongest parts of this paper. Previous work had recorded from the EP-LHb population in rodents and primates in head-fixed configurations, the recordings of this population in a freely moving context is a valuable addition to these studies and has highlighted more clearly that these neurons respond both at the time of choice and outcome.The use of a refined intersectional technique to record specifically the EP sst+ neurons is also an important strength of the paper. This is because previous work has shown that there are two genetically different types of glutamatergic EP neurons that project to the LHb. Previous work had not distinguished between these types in their recordings so the current results showing that the bidirectional value signaling is present in the EP sst+ population is valuable.Weaknesses:(1) One of the main weaknesses of the paper is to do with how the effect of the EP sst+ neurons on the behavior was assessed.(a) All the manipulations (blocking synaptic release and blocking glutamatergic transmission) are chronic and more importantly the mice are given weeks of training after the manipulation before the behavioral effect is assessed. This means that as the authors point out in their discussion the mice will have time to adjust to the behavioral manipulation and compensate for the manipulations. The results do show that mice can adapt to these chronic manipulations and that the EP sst+ are not required to perform the task. What is unclear is whether the mice have compensated for the loss of EP sst+ neurons and whether they play a role in the task under normal conditions. Acute manipulations or chronic manipulations without additional training would be needed to assess this.

Unfortunately, when mice are given a three week break from behavioral training (the time required to allow for adequate viral expression) behavioral performance on the task (*p*(highport), *p*(switch), trial number, trial time, etc.) is significantly degraded. Animals do eventually recover to previous performance levels, but this takes place during a 4-5 day “relearning” period. Here we sought to examine if EP^Sst+^ neurons are required for continued task performance and chose to continue to train the animals following viral injection to avoid the “relearning” period that occurs following an extended break from behavioral training which may have made it difficult to interpret changes in behavioral performance due to the viral manipulation vs relearning.

Acute manipulations were not used because we planned to compare complete synaptic ablation (Tettx) and single neurotransmitter ablation (CRISPR Slc17a6) over similar time courses and we know of no acute manipulation that could achieve single neurotransmitter ablation.

(b) Another weakness is that the effect of the manipulations was assessed in the 90/10 contingency version of the task. Under these contingencies, mice integrate past outcomes over fewer trials to determine their choice and animals act closer to a simple win-stay-lose switch strategy. Due to this, it is unclear if the EP sst+ neurons would play a role in the task when they must integrate over a larger number of conditions in the less deterministic 70/30 version of the task.

It is possible that a requirement for EP^Sst+^ neurons could be revealed if the experiment was conducted with different parameters (either different reward probabilities, fluctuating reward probabilities within a session, or withholding additional training during viral expression). It is difficult to predict which version of the task, if any, would be most likely to reveal a requirement for EP^Sst+^ neurons based on our results. We favor testing for EP^Sst+^ function using a new behavioral paradigm that allows us to carefully examine task learning following EP manipulations in an independent study.

The authors show an intriguing result that the EP sst+ neurons are excited when mice make an ipsilateral movement in the task either toward or away from the center port. This is referred to as a choice response, but it could be a movement response or related to the predicted value of a specific action. Recordings while mice perform movement outside the task or well-controlled value manipulations within the session would be needed to really refine what these responses are related to.

If activity of EP^Sst+^ neurons included a predicted value component, we would expect to see a change in activity during ipsilateral movements when the previous trial was rewarded vs unrewarded. This is examined in Fig 2—figure suppl. 2C, where we compare EP^Sst+^ responses during ipsilateral trials when the previous trials were either rewarded (blue) or unrewarded (gray). We show that EP^Sst+^ activity prior to side port entry (SE) is identical in these two trial types indicating that EP^Sst+^ neurons do not show evidence of predicted value of an action in this context. Therefore, we conclude that increased EP^Sst+^ activity during ipsilateral trials is primarily related to ipsilateral movement following CX (we call this the “choice” phase of the trial). We also show that other ipsiversive movements outside of the “choice” phase of a trial (such as the return to center port following a contralateral trial) show a smaller but significant increase in activity (Figure 2—figure supplement 1F-G). Therefore, whereas the activity observed during ipsilateral choice contains signals related to ipsilateral movement and additional factors, our data suggest that predicted value is not one of those factors. We will clarify this point and our definition of “choice” in the narrative.

(2) The authors conclude that they do not see any evidence for bidirectional prediction errors. It is not possible to conclude this. First, they see a large response in the EP sst+ neurons to the omission of an expected reward. This is what would be expected of a negative reward prediction error. There are much more specific well-controlled tests for this that are commonplace in head-fixed and freely moving paradigms that could be tested to probe this. The authors do look at the effect of previous trials on the response and do not see strong consistent results, but this is not a strong formal test of what would be expected of a prediction error, either a positive or negative. The other way they assess this is by looking at the size of the responses in different recording sessions with different reward contingencies. They claim that the size of the reward expectation and prediction error should scale with the different reward probabilities. If all the reward probabilities were present in the same session this should be true as lots of others have shown for RPE. Because however this data was taken from different sessions it is not expected that the responses should scale, this is because reward prediction errors have been shown to adaptively scale to cover the range of values on offer (Tobler et al., Science 2005). A better test of positive prediction error would be to give a larger-than-expected reward on a subset of trials. Either way, there is already evidence that responses reflect a negative prediction error in their data and more specific tests would be needed to formally rule in or out prediction error coding especially as previous recordings have shown it is present in previous primate and rodent recordings.

We do not conclude that we see no evidence for RPE and the reviewer is correct in stating that a large increase in EP^Sst+^ activity following omission of an expected reward would be expected of a negative reward prediction error. However, this observation alone is not strong enough evidence that EP^Sst+^ neurons signal RPE. When we looked for additional evidence of RPE within our experiments we did not find consistent demonstrations of its existence in our data. When performing photometry measurements of dopamine release in the striatum, RPE signals are readily observed with a task identical to ours using trial history to as a modifier of reward prediction (Chantranupong, et al 2023). Of course, there could be a weaker more heterogeneous RPE signal in EP^Sst+^ neurons that we cannot detect with our methods. As we state in the discussion, RPE signals may be present in a subset of individual neurons (as observed in Stephenson-Jones et al, 2016 and Hong and Hikosaka, 2008) which are below our detection threshold using fiber photometry. Additionally, Hong and Hikosaka, 2008 show that LHb-projecting GPi neurons show both positive and negative reward modulations which may obscure observation of RPE signals with photometry recordings that arise from population activity of genetically defined neurons.

(3) There are a lot of variables in the GLM that occur extremely close in time such as the entry and exit of a port. If two variables occur closely in time and are always correlated it will be difficult if not impossible for a regression model to assign weights accurately to each event. This is not a large issue, but it is misleading to have regression kernels for port entry and exits unless the authors can show these are separable due to behavioral jitter or a lack of correlation under specific conditions, which does not seem to be the case.

It is true that two variables that are always correlated are redundant in a GLM. For example, center entry (CE) and center exit (CX) occur in quick succession in most trials and are highly correlated (Figure 1C). For this reason, when only one is removed as a predictor from the model but not the other there is a very small change in the MSE of the fit (Figure 3E, *-CE or -CX*). However, when both are removed model performance decreases further indicating that center-port nose-pokes do contribute to model performance (Figure 3E, -*CE/CX*). Due to the presence/absence of reward following side port entry there is substantial behavioral jitter (due to water consumption in rewarded trials) that the SE and SX are not always correlated, therefore the model performs worse when either are omitted alone, but even worse still when both SE/SX are omitted together (Figure 3E, -*SE/SX*). We will update Figure 3 and the narrative to make this more explicit.

**Reviewer #3 (Public Review):**
Summary:The authors find that Sst-EPN neurons, which project to the lateral habenula, encode information about response directionality (left vs right) and outcome (rewarded vs unrewarded). Surprisingly, impairment of vesicular signaling in these neurons onto their LHb targets did not impair probabilistic choice behavior.Strengths:Strengths of the current work include extremely detailed and thorough analysis of data at all levels, not only of the physiological data but also an uncommonly thorough analysis of behavioral response patterns.Weaknesses:Overall, I saw very few weaknesses, with only two issues, both of which should be possible to address without new experiments:(1) The authors note that the neural response difference between rewarded and unrewarded trials is not an RPE, as it is not affected by reward probability. However, the authors also show the neural difference is partly driven by the rapid motoric withdrawal from the port. Since there is also a response component that remains different apart from this motoric difference (Figure 2, Supplementary Figure 1E), it seems this is what needs to be analyzed with respect to reward probability, to truly determine whether there is no RPE component. Was this done?

We thank the reviewer for this comment, we believe this is particularly important for unrewarded trials as SE and SX occur in rapid succession. In Figure 2—figure supplement 2A-B we now show the photometry signal from Rewarded and Unrewarded ipsilateral trials aligned to SX for different reward probabilities. We quantify the signals for different reward probabilities during a 500ms window immediately *prior* to SX but find no differences between groups.

(2) The current study reaches very different conclusions than a 2016 study by Stephenson-Jones and colleagues despite using a similar behavioral task to study the same Sst-EPN-LHb circuit. This is potentially very interesting, and the new findings likely shed important light on how this circuit really works. Hence, I would have liked to hear more of the authors' thoughts about possible explanations of the differences. I acknowledge that a full answer might not be possible, but in-depth elaboration would help the reader put the current findings in the context of the earlier work, and give a better sense of what work still needs to be done in the future to fully understand this circuit.For example, the authors suggest that the Sst-EPN-LHb circuit might be involved in initial learning, but play less of a role in well-trained animals, thereby explaining the lack of observed behavioral effect. However, it is my understanding that the probabilistic switching task forces animals to continually update learned contingencies, rendering this explanation somewhat less persuasive, at least not without further elaboration (e.g. maybe the authors think it plays a role before the animals learn to switch?).Also, as I understand it, the 2016 study used manipulations that likely impaired phasic activity patterns, e.g. precisely timed optogenetic activation/inhibition, and/or deletion of GABA/glutamate receptors. In contrast, the current study's manipulations - blockade of vesicle release using tetanus toxin or deletion of VGlut2, would likely have blocked both phasic and tonic activity patterns. Do the authors think this factor, or any others they are aware of, could be relevant?

We have added further discussion of the Stephenson-Jones, et al 2016 study as well as the Lazaridis, et al 2019 study which shows no effect of phasic stimulation of EP when specifically manipulating EP^Sst+^ (vGat+/vGlut2+) neurons rather than vGlut2+ neurons as in the Stephenson-Jones study.

**Recommendations for the authors:**

**Reviewer #1 (Recommendations For The Authors):**
In some places, there seems to be a mismatch between referenced figures and texts. For example:(1) The authors described that 'This increase in activity was seen for all three reward probabilities tested (90/10, 80/20, and 70/30) and occurred while the animal was engaged in ipsiversive movements as similar increases were observed following side exit (SX) on contralateral trials as the animal was moving from the contralateral side port back to the center port (Figure 2-Figure Supplement 1c)', but supplement 1c is not about calcium dynamics around the SX event. I presume they mean Figure 2-Figure Supplement 1d.

Yes, this will be corrected in the revised manuscript.

(2) The authors explained that increased EPSst+ neuronal activity following an unrewarded outcome was partially due to the rapid withdrawal of the animal's snout following an unrewarded outcome however, differences in rewarded and unrewarded trials were still distinguishable when signals were aligned to side port exit indicating that these increases in EPSst+ neuronal activity on unrewarded trials were a combination of outcome evaluation (unrewarded) and side port withdrawal occurring in quick succession (SX, Figure 2 - Figure Supplement 1d). I presume that they mean Figure 2 - Figure Supplement 1e.

Yes, this will be corrected in the revised manuscript.

Minor suggestions related to specific figure presentation are below:Figure 2 and supplement figures:(1) Figure 2B: the authors may consider presenting outcome-related signals recorded from all trials, including both ipsilateral and contralateral events, and align signals to SE when reward consumption presumably begins, rather than aligning to CE.

We have added sample recordings from ipsilateral and contralateral trials and sorted them by trial duration to allow for clearer presentation of activity following CE and SE (Figure 2—figure supplement 1a-b).

(2) The authors described that 'This increase in activity was seen for all three reward probabilities tested (90/10, 80/20, and 70/30) and occurred while the animal was engaged in ipsiversive movements as similar increases were observed following side exit (SX) on contralateral trials as the animal was moving from the contralateral side port back to the center port (Figure 2-Figure Supplement 1c)', but supplement 1c is not about calcium dynamics around the SX event. I presume they mean Figure 2-Figure Supplement 1d.

Yes, this will be corrected in the revised manuscript.

(3) The authors explained that increased EPSst+ neuronal activity following an unrewarded outcome was partially due to the rapid withdrawal of the animal's snout following an unrewarded outcome however, differences in rewarded and unrewarded trials were still distinguishable when signals were aligned to side port exit indicating that these increases in EPSst+ neuronal activity on unrewarded trials were a combination of outcome evaluation (unrewarded) and side port withdrawal occurring in quick succession (SX, Figure 2 -Figure Supplement 1d). I presume that they mean Figure 2 -Figure Supplement 1e.

Yes, this will be corrected in the revised manuscript.

Figure 3 and supplement figures:(1) Figure 3C-F: it is hard to compare the amplitude of calcium signals between different behaviour events without a uniform y-axis.

The scale for the y-axis on Figure 3C-D is uniform for all panels. Figure 3E is also uniform for all boxplots. The reviewer may be referring to Figure 2C-F, but the y-axis for all of the photometry data is uniform for all panels and the horizontal line represents zero. The y-axis for the quantification on the right of each panel is scaled to the max/min for each comparison.

(2) Figure 3E is difficult to follow. The authors explained that the 'SE' variable is generated by collapsing the ipsilateral and contralateral port entries, and hence the variable has no choice of direction information. I assumed that the 'SX', 'CE', and 'CX' variables are generated similarly. It is not clear if this is the case for the 'side', 'centre' and 'choice' variables. The authors explained that 'omitting center port entry/exit together or individually also resulted in decreased GLM performance but to a smaller degree than the omission of choice direction (Figure 3e, "-Center")'. My understanding is that they created the Centre variable by collapsing ipsilateral and contralateral centre port entry/exit together. The Centre variable should have no choice of direction information. How is the Center variable generated differently from omitting centre port entry/exit together? I would ask the authors to explain the model and different variables a bit more thoroughly in the text.

We apologize for the confusion. All ten variables used to train the full GLM are listed in Fig. 3C. In Figure 3E variable(s) were omitted to test how they contributed to GLM performance (data labeled “None” is the full model with all variables). Omitted variables are now defined as follows: *-Rew* = Rew+Unrew removed, *-Direction* = Ipsi/Contra designation removed and collapsed into CE, CX, SE, SX, *-Direction & Rew* = Ipsi/Contra info removed from all variables + Rew/Unrew removed, *-CE/CX* = Ipsi/Contra CE and CX removed, *-CE* = Ipsi/contra CE removed, *-CX* = Ipsi/contra CX removed, *-SE/SX* = Ipsi/Contra SE and SX removed, *-SE* = Ipsi/contra SE removed, *-SX* = Ipsi/contra SX removed. This clarification has also been added to the *Generalized Linear Model* section of Materials and Methods.

Figure 5 and supplement figures:There are no representative and summary figures show the specificity and efficiency of oChief-tdTomato or Tetx-GFP expression. Body weight changes following virus injection are not well described.

A representative image of Tettx GFP expression are shown in Fig. 4A and percent of infected EP^Sst+^ neurons is described in the text (70±15.1% (mean± SD), 1070±230 neurons/animal, n=6 mice). Most oChief-tdTom animals were used for post-hoc electrophysiology experiments and careful quantification of viral expression was not possible. However, *Slc17a6* deletion was confirmed in these animals (Fig. 5 – Fig supplement 1J-K) to confirm the manipulation was effective in the experimental group. A representative image of oChief-tdTom expression is shown in Fig. 5A.

We now mention the body weight changes observed following Tettx injection in the narrative.

**Reviewer #2 (Recommendations For The Authors):**
(1) In the RFLR section you state that "this variable decays...", a variable can't decay only the value of a variable can change. Also, it is not mentioned what variable is being discussed. There are lots of variables in the model so this should be made clear.

We now state, “This variable (β) changes over trials and is updated with new evidence from each new trial’s choice and outcome with an additional bias towards or away from its most recent choice (Figure 1-figure supplement 2A-C).”

(2) I couldn't find in the results section, or the methods section the details for the Tet tx experiments, were mice trained and tested on 90/10 only? Were they trained while the virus was expressing etc? This should be added.

In the methods section we state, ”For experiments where we manipulated synaptic release in EP*Sst+* neurons (Figures 4-5) we trained mice (reward probabilities 90/10, no transparent barrier present) to the following criteria for the 5 days prior to virus injection: (1) *p*(highport) per session was greater than or equal to 0.80 with a variance less than 0.003, (2) *p*(switch) per session was less than or equal to 0.15 with a variance less than 0.001, (3) the *p*(left port) was between 0.45-0.55 with a variance less than 0.005, and (4) the animal performed at least 200 trials in a session. The mean and variance for these measurements was calculated across the five session immediately preceding surgery. The criterion were determined by comparing performance profiles in separate animals and chosen based on when animals first showed stable and plateaued behavioral performance. Following surgery, mice were allowed to recover for 3 days and then continued to train for 3 weeks during viral expression. Data collected during the 5 day pre-surgery period was then compared to data collected for 10 sessions following the 3 weeks allotted for viral expression (i.e. days 22-31 post-surgery).”

**Reviewer #3 (Recommendations For The Authors):**
(1) The kernel in Figure 3C shows an activation prior to CE on "contra" trials that is not apparent in Figure 2C which shows no activation prior to CE on either contra or ipsi trials. Given that movement directionality prior to CE is dictated by the choice on the PREVIOUS trial, is the "contra" condition in 3C actually based on the previous trial? If so, this should be clarified.

On most “contra” trials the animal is making an ipsiversive movement just prior to CE as it returns to the center from the contralateral side-port (as most trials are no “switch” trials). Therefore, an increase in activity is expected and shown most clearly following SX for contralateral trials in Fig 2 –Fig suppl 1F. A significant increase in activity prior to CE on contra trials compared to ipsi trials can also be seen in Fig 2C, its just not as large a change as the increase observed following CE for ipsi. trials. The comparison between activity observed during the two types of ipsiversive movements is now shown directly in Figure 2—figure supplement 1G.

(2) Paragraph 7 of the discussion uses a phrase "by-in-large", which probably should be "by and large".

Thank you for the correction.

**Editor's note:**
Should you choose to revise your manuscript, if you have not already done so, please include full statistical reporting including exact p-values wherever possible alongside the summary statistics (test statistic and df) and 95% confidence intervals. These should be reported for all key questions and not only when the p-value is less than 0.05 in the main manuscript.Readers would also benefit from coding individual data points by sex and noting N/sex.

Sex breakdown has been added to figure legends for each experiment, full statistical reporting is now also include in the figure legends.